# LEARNING FLEXIBLE FORWARD TRAJECTORIES FOR MASKED MOLECULAR DIFFUSION

**Hyunjin Seo[1,2]\*, Taewon Kim[1,2]\*, Sihyun Yu[1], Sungsoo Ahn[1]†**
Korea Advanced Institute of Science and Technology (KAIST)[1], Polymerize[2]
{bella72, maxkim139, sihyun.yu, sungsoo.ahn}@kaist.ac.kr

## ABSTRACT

Masked diffusion models (MDMs) have achieved notable progress in modeling discrete data, while their potential in molecular generation remains underexplored. In this work, we explore their potential and introduce the surprising result that naïvely applying standards MDMs to molecules leads to severe performance degradation. We trace this critical issue to a *state-clashing problem*—where the forward diffusion trajectories of distinct molecules collapse into a common state, resulting in a mixture of reconstruction targets that cannot be learned with a typical reverse diffusion with unimodal predictions. To mitigate this, we propose **M**asked **E**lement-wise **L**earnable **D**iffusion (**MELD**) that orchestrates per-element corruption trajectories to avoid collisions between different molecular graphs. This is realized through a parameterized noise scheduling network that learns distinct corruption rates for individual graph elements, *i.e.*, atoms and bonds. Across extensive experiments, **MELD** achieves 100% chemical validity in unconditional generation on QM9 and ZINC250K datasets, while markedly improving distributional and property alignment over standard MDMs on both conditional and unconditioned generation. **Project page:** https://holymollyhao.github.io/MELD

## 1 INTRODUCTION

Molecular generation is critical in a variety of real-world applications, such as drug discovery (Simonovsky & Komodakis, 2018) and material design (Jia et al., 2024; Yang et al., 2023). However, the task remains challenging due to the extremely large and complex nature of the chemical space (Du et al., 2024). With the remarkable recent progress in deep generative models (Kingma & Welling, 2013; Rezende & Mohamed, 2015; Austin et al., 2021; Naveed et al., 2023), many approaches have attempted to tackle this problem by training a neural network that learns molecular distributions from large molecular datasets, demonstrating a strong promise in accelerating molecule discovery (Jensen, 2019; Jin et al., 2018; Shi et al., 2020; Jo et al., 2022; Vignac et al., 2023; Yiming et al., 2025).

In particular, recent works have focused on exploring generative models based on denoising diffusion or flow-matching models, (Jo et al., 2022; Lee et al., 2023; Vignac et al., 2023; Kong et al., 2023; Jo et al., 2024; Liu et al., 2024a), to learn a molecular distribution, inspired by their great success in other data domains with scalability (Ho et al., 2020; Song et al., 2020; Austin et al., 2021; Nichol & Dhariwal, 2021; Ma et al., 2024; Kingma et al., 2021; Sahoo et al., 2024b; Wan et al., 2025). These models learn to recover original molecules from corrupted versions through a denoising process, where the corruption typically involves altering types of atoms and bonds (*e.g.*, changing a carbon atom to nitrogen, or a single bond to a double bond).

Meanwhile, researchers have explored masked diffusion models (MDMs; Austin et al. 2021; Chang et al. 2022; Shi et al. 2024; Sahoo et al. 2024a). Unlike conventional diffusion models that typically design diffusion processes in continuous space, MDMs are specialized for discrete data by defining a diffusion process more suitable in discrete space. Specifically, MDMs define the forward process as element masking and train the model to infill the masked element during the reverse process. Intriguingly, MDMs show great stability and scalability, being comparable or even better than previous

---

*Equal Contribution.
†Corresponding Author.

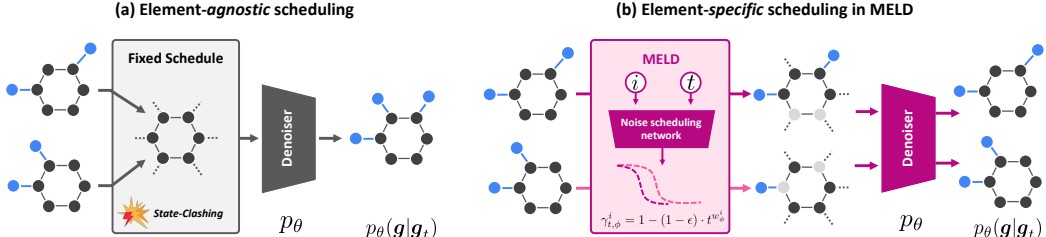

Figure 1: Comparison between (a) element-agnostic noise scheduling and (b) element-specific noise scheduling. The former results in an issue denoted as *state-clashing*, leading to generation of invalid molecules. **MELD** mitigates this with element-specific noise schedule, effectively orchestrating the forward process to minimize state-clashings.

generative models for discrete data, such as autoregressive language models (Ziegler & Rush, 2019; Hoogeboom et al., 2021) or high-resolution text-to-image generation (Chang et al., 2023). Despite their success in other domains, applying MDMs to molecular graphs is still underexplored.

In this work, we focus on applying MDMs to molecular generation. Surprisingly, unlike other domains, a naïve adaptation of existing MDMs to molecular graphs results in significantly worse performance, often generating distributionally misaligned structures. We argue that this phenomenon stems from a *state-clashing problem*: Molecular graphs with different properties and semantics easily collapse into a common intermediate state in the forward process (see Figure 1(a) for an illustration). We attribute this to the usage of fixed noise schedules; element-agnostic masking rates across all nodes and edges. This hinders the learning process of the unimodal denoiser – which predicts nodes and edges independently, by creating multimodal targets for reconstruction. Such mismatch forces the model to spread its probability mass into an averaged state creating samples that deviate from the true target distribution and, in some cases, violate chemical rules.

To address this, we introduce **MELD** (**M**asked **E**lement-wise **L**earnable **D**iffusion), a novel MDM for molecular graph generation. The main idea of our method is to alleviate the state-clashing problem by proposing an *element-wise learnable forward process*, which generates corruption trajectories in the way of minimizing the occurrence of potential collision. To this end, we introduce a parameterized noise scheduling network to yield distinct corruption rates for individual graph elements (*i.e.*, for nodes or edges). During training, we jointly optimize the forward (*i.e.*, noise scheduling network) and the reverse process (*i.e.*, MDM denoiser network). Intuitively, by assigning per-element trajectories, **MELD** organizes the forward process such that the probability of molecules being collapsed to the same intermediate state (see Figure 1(b)) is minimized). Through such evasion, **MELD** effectively learns to produce samples capturing the target molecular distribution.

We evaluate **MELD** on diverse molecular datasets, including QM9 (Ramakrishnan et al., 2014), Polymers (Thornton et al., 2012), ZINC250K (Irwin et al., 2012), Guacamol (Brown et al., 2019), and a synthetic graph benchmark (Martinkus et al., 2022). First, we demonstrate that **MELD** yields substantial improvements in distributional similarity over standard MDMs, while maintaining 100% validity. In conditional generation, **MELD** further enhances property alignment by up to 13.4% over state-of-the-art baseline. Finally, we show the scalability and generalizability of **MELD** in large-scale molecules and non-molecule graph datasets.

Our contributions are threefold:

- We identify a key limitation in applying standard masked diffusion models to molecular generation, the use of an element-agnostic noise schedule, which leads to frequent *state-clashing*.
- We present **MELD**, a novel masked diffusion framework that mitigates the state-clashing problem by learning per-element noise schedules, allowing adaptive corruption trajectories tailored to individual molecular components.
- **MELD** substantially improves the overall quality of generated molecules over standard MDM baselines, and surpasses existing molecular diffusion models in both unconditional and property-conditioned generation tasks. Moreover, its efficacy generalizes consistently to large-scale molecule and synthetic graph benchmarks.

## 2 RELATED WORK

**Masked diffusion models (MDMs).** MDMs have emerged as a powerful generative modeling scheme for discrete data generation. Initially, D3PM (Austin et al., 2021) introduces an absorbing mask token into the forward process and establishes a conceptual bridge between discrete diffusion and masked language modeling. Additionally, in image generation, MaskGIT (Chang et al., 2022) shows that generative modeling based on unmasking enables fast and qualitatively comparable high-fidelity image synthesis compared with left-to-right autoregressive decoding. More recent efforts have further refined MDMs to close the performance gap with autoregressive models (AR; Vaswani et al. 2017; Ziegler & Rush 2019; Hoogeboom et al. 2021). Notably, MD4 (Shi et al., 2024) and MDLM (Sahoo et al., 2024a) show that the diffusion objective can be simplified as a weighted integral of cross-entropy and that the model can achieve state-of-the-art results over prior diffusion models.

However, naive adoption of the MDM framework in molecular graph generation introduces unique challenges, termed as *state-clashing problem*. As molecular graphs exhibit higher symmetries while utilizing smaller vocabulary, the forward process of MDMs easily collapse distinct graphs into a same intermediate state, hindering the learning process, as evidenced in Tables 1 and 2. We formulate this problem further in Section 4.1.

**Diffusion models for molecules.** The success of diffusion models for image (Rombach et al., 2022) and text generation (Li et al., 2022) has inspired researchers to explore diffusion models in molecule domain. A surge of studies (Vignac et al., 2023; Jo et al., 2022; Xie et al., 2021; Kong et al., 2023; Jo et al., 2024; Liu et al., 2024a) have been proposed to generate de novo molecules, competing with sequential models (Segler et al., 2018; Jin et al., 2018; Shi et al., 2020; Jang et al., 2024a;b) that iteratively constructs a graph by adding graph elements. These efforts can be categorized into two approaches: (1) Score-based molecule diffusion approaches (Jo et al., 2022; Lee et al., 2023; Jo et al., 2024) adopt continuous noise on molecular graphs using stochastic differential equations (SDEs) (Song et al., 2020). They train a score function to approximate reverse SDEs, relaxing discrete atoms/bonds into a continuous space. (2) Discrete diffusion-based approaches (Vignac et al., 2023; Liu et al., 2024a; Hua et al., 2024; Kerby & Moon, 2024) apply discrete noise through Markovian transitions to nodes and edges in molecular graphs. Then they train a denoising neural network to reconstruct perturbed atom and bond types.

Despite progress in these two directions, masked diffusion frameworks remain underexplored for molecular generation. A preliminary application was explored in Kong et al. (2023), but it generates atoms in an autoregressive manner, limiting its ability to exploit the parallelized reconstruction of MDMs. In contrast, we propose MDMs for molecular graphs by focusing on the state-clashing problem occurring in the forward process, while preserving the parallelism inherent to MDMs.

## 3 PRELIMINARIES

We provide a brief overview of masked diffusion models for molecular generation. The goal is to generate molecular graphs $\boldsymbol{g} = (\boldsymbol{x}, \boldsymbol{e})$ from a data distribution $q(\boldsymbol{g})$, where $\boldsymbol{x} = (x^i)_{i=1}^N$ and $\boldsymbol{e} = (e^{ij})_{i,j=1}^N$ represent one-hot encoded node and edge features, each augmented with an absorbing [mask] token. Following standard diffusion frameworks (Ho et al., 2020; Ma et al., 2024; Nichol & Dhariwal, 2021; Peebles & Xie, 2023), we consider a forward process $q_\phi(\boldsymbol{g}_t|\boldsymbol{g}_{t-1})$ and a reverse process $p_\theta(\boldsymbol{g}_{t-1}|\boldsymbol{g}_t)$, parameterized by $\phi$ and $\theta$, respectively.

The forward process is defined as follows, where $\gamma_{t,\phi}$ denotes the marginal masking probabilities parameterized by $\phi$:

$$q_\phi(x_t^i \,|\, x_0^i) = \begin{cases} \gamma_{t,\phi}^i & \text{if } x_t^i = [\text{mask}] \\ 1 - \gamma_{t,\phi}^i & \text{if } x_t^i = x_0^i \end{cases}, \quad q_\phi(e_t^{ij} \,|\, e_0^{ij}) = \begin{cases} \gamma_{t,\phi}^{ij} & \text{if } e_t^{ij} = [\text{mask}] \\ 1 - \gamma_{t,\phi}^{ij} & \text{if } e_t^{ij} = e_0^{ij} \end{cases} \quad (1)$$

Most existing molecular diffusion models (Vignac et al., 2023; Jo et al., 2022; Lee et al., 2023; Jo et al., 2024; Liu et al., 2024a) have defined the corruption probability using a *fixed, element-agnostic* noise scheduling function (*i.e.*, $\gamma_t$).

The denoiser predicts the original graph $\boldsymbol{g}_0$ by independently predicting nodes and edges. It is trained to recover the original graph directly without recursive sampling (Vignac et al., 2023; Liu et al.,

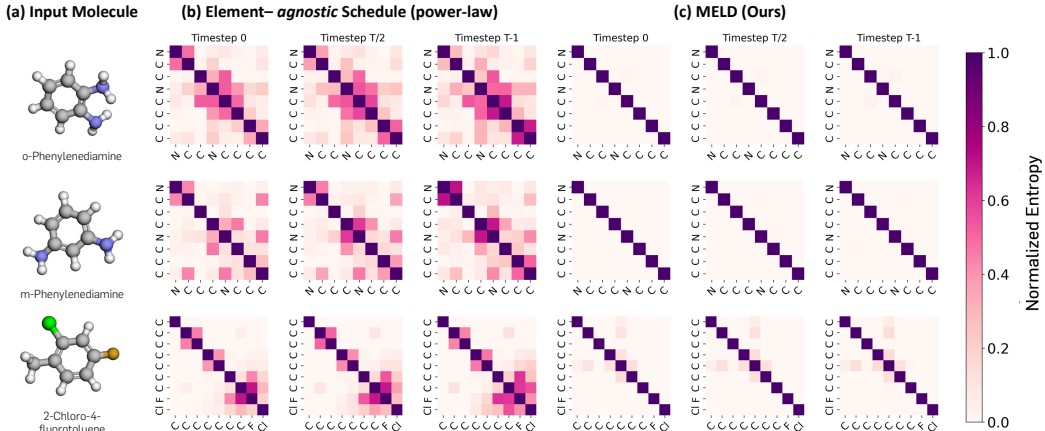

Figure 2: Visualization of prediction entropy for various molecule types. The first and second rows show prediction matrices with nitrogen bonds masked, while the third row shows generations with chlorine bond masked. From left to right: (a) 3D renderings of the input molecules, (b) predictions from MDMs using a fixed power law noise schedule, and (c) predictions from **MELD**. Brighter colors indicate lower uncertainty (*i.e.*, higher confidence). The dark diagonal entries reflect enforced uniform predictions, as self-connections in molecules are not meaningful and are excluded from valid outputs. Note that predictions are being made for all locations, regardless of their entropy values.

2024a), by minimizing the following loss objective:

$$\mathcal{L}(\theta, \phi) = \mathbb{E}_{t,\boldsymbol{g},\boldsymbol{g}_t} \left[ \sum_{1 \leq i \leq N} \frac{\dot{\gamma}^i_{t,\phi}}{1 - \gamma^i_{t,\phi}} \log p_\theta(x^i | \boldsymbol{g}_t) + \lambda \sum_{1 \leq i < j \leq N} \frac{\dot{\gamma}^{ij}_{t,\phi}}{1 - \gamma^{ij}_{t,\phi}} \log p_\theta(e^{ij} | \boldsymbol{g}_t) \right] \quad (2)$$

Here, $\dot{\gamma}_{t,\phi}$ denotes the derivative of $\gamma_{t,\phi}$ with respect to $t$, while $\lambda > 0$ balances node- and edge-level reconstruction, following prior work (Vignac et al., 2023; Liu et al., 2024a).

# 4 MELD: MASKED ELEMENT-WISE LEARNABLE DIFFUSION

In this section, we introduce **MELD**, a masked diffusion model (MDM) for molecular graph generation that jointly learns per-graph-element corruption rate and the denoising model. As we will explain, our proposed design alleviates the state-clashing problem (Section 4.1) by producing distinguishable forward trajectories for each molecular component (Section 4.2).

## 4.1 FORMALIZING THE STATE-CLASHING PROBLEM

In this section, we describe the *state-clashing problem* which naturally arise for training MDMs on graphs without learning the forward process, *i.e.*, set $\gamma^i_{t,\phi}$ to some constant $\gamma_t$ for all node $i$ and edges $(i, j)$. In a nutshell, state-clashing refers to the phenomenon where semantically distinct molecules are corrupted into the same intermediate state, due to the nature of the constant forward process in MDMs. Consequently, the model trained with such constant forward process struggles to infer the correct reconstruction target, resulting in outputs that fail to preserve structural or molecular coherence with target distribution (see Figure 1 for an illustration). This is particularly pronounced in molecules with symmetric motifs, to which the number of immediate parent states grows by the number of permutations that leave the motif invariant.

Formally, note that the diffusion model loss in Equation 2 can be expressed as:

$$\mathbb{E}_t \left[ \mathrm{KL}(p(\boldsymbol{g}|\boldsymbol{g}_t), p_\theta(\boldsymbol{g}|\boldsymbol{g}_t)) \right], \quad p(\boldsymbol{g}|\boldsymbol{g}_t) \propto p(\boldsymbol{g}_t|\boldsymbol{g})p(\boldsymbol{g}). \quad (3)$$

The main problem is that $p(\boldsymbol{g}|\boldsymbol{g}_t)$ can be highly *multimodal*, *i.e.*, there exists many graph $\boldsymbol{g}$ with non-zero probabilities of $p(\boldsymbol{g}_t|\boldsymbol{g})$. However, the parameterized diffusion model $p_\theta(\boldsymbol{g}|\boldsymbol{g}_t) = \prod_{1 \leq i \leq N} p_\theta(x^i|\boldsymbol{g}_t) \prod_{1 \leq i < j \leq N} p_\theta(e^{ij}|\boldsymbol{g}_t)$ is *unimodal*, as it predicts each node and edge independently, typically resulting in a single mode centered around an average graph. Furthermore, due to the mode-covering property of KL divergence, the reverse diffusion model trained with Equation 2

tends to converge to a high-entropy distribution–the model compensates for its inability to represent multiple modes by spreading its probability mass broadly around the single mode.

This is illustrated in Figure 2, where we visualize the denoiser's prediction entropy when reconstructing masked bonds in the given molecules. In the first two rows, we mask all nitrogen–carbon bonds in o- and m-phenylenediamine. As masking removes the distinguishing nitrogen atoms, both molecules collapse into the identical symmetric benzene backbone, creating a severe state-clashing scenario. Under an element-agnostic schedule, the denoiser exhibits higher uncertainty when predicting the masked bond types, as many distinct underlying configurations are compatible with the same corrupted state. Additionally, we visualize the denoiser prediction for 2-Chloro-4-fluorotoluene when only the chlorine bond is masked. Due to the inherent asymmetry of the masked molecule, the state-clashing issue is less pertinent than Phenylenediamine isomers. Consequently, the denoiser shows increased prediction confidence even with element-agnostic schedules, underscoring the necessity of addressing the state-clashing.

We note that this issue is not unique to MDMs, but it does become significantly more severe in their case. Masking operations tend to absorb diverse input graphs into indistinguishable intermediate states, whereas the probability of state-clashing in substitution-based discrete diffusion is orders of magnitude lower with realistic vocabulary sizes (see Appendix **??** for a detailed qualitative analysis). Moreover, the effect is particularly pronounced in molecular graphs, which often contain structural symmetries and a limited set of element types compared to other discrete domains.

## 4.2 MAIN ALGORITHM

**Learnable element-wise embedding.** To reduce state-clashing in forward diffusion trajectories across graph states, one should use information that distinguishes individual graph elements, which guides the noise scheduling network. One can consider incorporating graph positional encodings (Dwivedi et al., 2022; Ma et al., 2023) for conditioning. However, such encodings often fail to disambiguate elements when given symmetric structures such as those found in aromatic rings (Lawrence et al., 2025; Morris et al., 2024). Moreover, conditioning the noisy graph input in the noise schedule breaks the tractable closed-form marginal $q(\boldsymbol{g}_t \mid \boldsymbol{g}_0)$ since the transition kernel becomes dependent on the current corrupted state, which eliminates the efficiency.

Thus, we consider learnable element-wise embeddings over the graph elements that assigns distinct masking rate, and use it for an input to the noise scheduling network. Specifically, we assign a learnable embedding matrix $\boldsymbol{H} \in \mathbb{R}^{D \times N}$ and consider its $i$-th column $\boldsymbol{h}^i$ as node-wise embedding of $i$-th node $x^i$, where $N > 0$ is a number of nodes and $D$ is the embedding dimension. For an edge $\{i, j\} \in \mathcal{E}$, we set its embedding by $\boldsymbol{h}^{ij} = \boldsymbol{h}^i + \boldsymbol{h}^j$. In addition, we randomly permute columns of $\boldsymbol{H}$ during training to differentiate graph states that have the same numbers of nodes and edges.

**Time-dependent noise schedule.** We parameterize the noise scheduling network for each element (*e.g.*, node) using a power-law function, commonly used in Shi et al. (2024; 2025). Leveraging $i$-th node embedding $\boldsymbol{h}^i$ as an example, our noise schedule $\gamma_{t,\phi}^i$ is defined as:

$$\gamma_{t,\phi}^i = 1 - (1 - \epsilon) \cdot t^{w_\phi^i}, \quad w_\phi^i = \sigma_{\text{sf}}(f_\phi(\boldsymbol{h}^i)), \tag{4}$$

where $\sigma_{\text{sf}}$ denotes the softplus function and $f_\phi(\cdot)$ is a linear layer. The same computation applies analogously to other nodes and edges. Consistent with Shi et al. (2024; 2025), we introduce a bounding constant $\epsilon$ for numerical stability and fix $\epsilon = 0.0001$ in all experiments. Throughout this process, **MELD** naturally introduces element-specific masking rates, mitigating the collapse between distinct molecules that would otherwise persist under element-agnostic noise scheduling.

**Maintaining gradient flow in discrete sampling.** In discrete-space molecular diffusion frameworks (Vignac et al., 2023; Liu et al., 2024a; Kerby & Moon, 2024), the noisy graph at each timestep is obtained by sampling a single graph from a categorical distribution over nodes and edges (Equation 1), as computing the full expectation over $\boldsymbol{g}_t \sim q(\cdot|\boldsymbol{g})$ is intractable. However, such discretization introduces a discontinuity in the computational graph when parameterizing the forward process, impeding a gradient flow towards the learnable schedule parameters $\phi$. Thus, we adopt the Straight-Through Gumbel-Softmax (STGS) estimator (Jang et al., 2017), which provides a differentiable surrogate for discrete sampling. This formulation ensures the forward pass to utilize one-hot vectors

for graph constructions, while the backward pass approximates them as continuous variables to enable end-to-end training.

Let $\boldsymbol{z} \in \mathbb{R}^N$ denote the logits, and $\eta > 0$ be the temperature parameter. We first compute a soft approximation of the categorical distribution $\boldsymbol{p}_{\text{soft}} \in [0, 1]^N$ via the Gumbel-Softmax:

$$\boldsymbol{p}_{\text{soft},k} = \frac{\exp((z_k + g_k)/\eta)}{\sum_{l=1}^{N} \exp((z_l + g_l)/\eta)}, \tag{5}$$

where $g_k = -\log(-\log(u_k))$ is a gumbel noise with $u_k \sim \text{Unif}[0, 1]$ and $z_k$ is the $k$-th element of the logits $\boldsymbol{z}$. A discrete one-hot vector $\boldsymbol{p}_{\text{hard}} \in \{0, 1\}^N$ is then obtained by taking the index with the highest probability:

$$k^* = \arg\max_k \boldsymbol{p}_{\text{soft},k}, \quad \boldsymbol{p}_{\text{hard},k} = \begin{cases} 1 & \text{if } k = k^* \\ 0 & \text{otherwise} \end{cases} \tag{6}$$

To retain gradient flow, we use the straight-through estimator to combine the discrete and continuous components, *i.e.*, set $\boldsymbol{p} = \boldsymbol{p}_{\text{hard}} - \text{sg}(\boldsymbol{p}_{\text{soft}}) + \boldsymbol{p}_{\text{soft}}$, where $\text{sg}(\cdot)$ denotes the stop-gradient operator. This ensures that the forward pass uses the discretized one-hot vector $\boldsymbol{p} = \boldsymbol{p}_{\text{hard}}$, while the backward pass treats $\boldsymbol{p}$ as the continuous $\boldsymbol{p}_{\text{soft}}$, allowing gradients to propagate through $z$, *i.e.*, $\frac{\partial \boldsymbol{p}}{\partial z} = \frac{\partial \boldsymbol{p}_{\text{soft}}}{\partial z}$.

**Permutation equivariance.** While strict equivariance is essential for molecular representation learning, *e.g.*, mapping isomorphic graphs to identical embeddings, graph generation only requires the learned distribution to be permutation invariant. Our method achieves this invariance not by constraining the architecture, but by marginalizing over permutations. Formally, we model the probability of a graph $\boldsymbol{g}$ as the expectation over all possible node orderings $\pi$, *i.e*, $p(\boldsymbol{g}) = \sum_\pi p(\boldsymbol{g}, \pi)$, where the summation is over the possible permutation $\pi$. Our randomized permutation strategy corresponds to maximizing the evidence lower-bound (ELBO) of this marginal log-likelihood:

$$\log p(\boldsymbol{g}) = \log \sum_\pi p(\boldsymbol{g}, \pi) \geq \mathbb{E}_\pi[\log p(\boldsymbol{g} \mid \pi)] + \text{const.} \tag{7}$$

This stochastic symmetrization is well-established paradigm in graph generation, widely used in autoregressive models (You et al., 2018b; Kong et al., 2023), which are inherently node-order dependent yet generate valid, invariant graph distributions.

**Domain specialization and applicability.** In principle, **MELD** is applicable to non-molecular data. However, we note that other discrete data such as text or protein sequences typically involve larger vocabularies and fewer structural symmetries. Consequently, the risk of collapsing distinct inputs into identical intermediate states is lower, and the relative benefits of **MELD** may be less pronounced in such settings. Nevertheless, to show the generality of our approach, we include additional experiments on general graph with constrained number of nodes and edges in Section 5.4.

## 5 EXPERIMENTS

### 5.1 EXPERIMENTAL SETUP

We evaluate **MELD** on unconditional and property-conditioned molecular generation tasks. For unconditional generation, in line with prior work (Jo et al., 2024; Kong et al., 2023; Jo et al., 2022), we use QM9 (Ramakrishnan et al., 2014), ZINC250k (Irwin et al., 2012), and Guacamol (Brown et al., 2019) datasets. For conditional generation, we adopt the Polymer dataset (Thornton et al., 2012) introduced in Liu et al. (2024a), which conditions homopolymers on three gas permeability constraints and synthesizability scores. We compare against recent baselines with standard metrics for both tasks, following established setups (Liu et al., 2024a; Jo et al., 2022; 2024). See Section C for detailed description of each method and metrics. Our implementation employs the diffusion transformer (Peebles & Xie, 2023) as the denoising network within a masked diffusion framework. For property-conditioned generation, we further apply classifier-free guidance (Ho & Salimans, 2021) as implemented in (Peebles & Xie, 2023; Liu et al., 2024a). Unless otherwise noted, all experiments use the same backbone across standard MDMs and **MELD**.

Table 1: Unconditional generation of 10K molecules on QM9 and ZINC250K datasets. The best and second best performances are represented by **bold** and underline.

| | QM9 | | | | | | ZINC250K | | | | | |
|---|---|---|---|---|---|---|---|---|---|---|---|---|
| Method | Valid.↑ | FCD↓ | NSPDK↓ | Scaf.↑ | Uniq.↑ | Novel.↑ | Valid.↑ | FCD↓ | NSPDK↓ | Scaf.↑ | Uniq.↑ | Novel.↑ |
| *Flow-based* | | | | | | | | | | | | |
| MoFlow | 91.36 | 4.47 | 0.017 | 0.145 | 98.65 | 94.72 | 63.11 | 20.93 | 0.046 | 0.013 | 99.99 | **100.00** |
| GraphAF | 74.43 | 5.63 | 0.021 | 0.305 | 88.64 | 86.59 | 68.47 | 16.02 | 0.044 | 0.067 | 98.64 | 99.99 |
| GraphDF | 93.88 | 10.93 | 0.064 | 0.098 | 98.58 | **98.54** | 90.61 | 33.55 | 0.177 | 0.000 | 99.63 | **100.00** |
| *Continuous diffusion* | | | | | | | | | | | | |
| EDP-GNN | 47.52 | 2.68 | 0.005 | 0.327 | **99.25** | 86.58 | 82.97 | 16.74 | 0.049 | 0.000 | 99.79 | **100.00** |
| GDSS | 95.72 | 2.90 | 0.003 | 0.698 | 98.46 | 86.27 | 97.01 | 14.66 | 0.019 | 0.047 | 99.64 | **100.00** |
| GruM | 99.69 | 0.11 | **0.0002** | 0.945 | 96.90 | 24.15 | 98.65 | 2.26 | 0.0015 | 0.530 | 99.97 | 99.98 |
| *Discrete diffusion* | | | | | | | | | | | | |
| DiGress | 98.19 | 0.10 | 0.0003 | 0.936 | 96.67 | 25.58 | 94.99 | 3.48 | 0.0021 | 0.416 | 99.97 | 99.99 |
| *Masked diffusion* | | | | | | | | | | | | |
| GraphARM | 90.25 | 1.22 | 0.002 | N/A | 95.62 | 70.39 | 88.23 | 16.26 | 0.055 | N/A | 99.46 | **100.00** |
| MDM w/ cosine | **100.00** | 3.67 | 0.009 | 0.653 | 85.96 | 69.85 | **100.00** | 25.41 | 0.051 | 0.001 | 99.99 | **100.00** |
| MDM w/ polynomial | **100.00** | 3.70 | 0.010 | 0.890 | 86.57 | 67.18 | **100.00** | 26.43 | 0.053 | 0.001 | 99.93 | **100.00** |
| MDM w/ power-law | **100.00** | 3.62 | 0.007 | 0.628 | 91.30 | 76.65 | **100.00** | 26.09 | 0.068 | 0.001 | **100.00** | **100.00** |
| **MELD (Ours)** | **100.00** | **0.09** | **0.0002** | **0.947** | 96.49 | 33.55 | **100.00** | **1.51** | **0.0006** | **0.559** | **100.00** | 99.96 |

Table 2: Property-conditioned generation of 10K Polymers on three gas permeability properties and synthetic score. The numbers in parentheses in Valid. represent the validity without correction. The best and second best performances are represented by **bold** and underline.

| | General Quality | | | | Property Alignment | | | | |
|---|---|---|---|---|---|---|---|---|---|
| Method | Valid.↑ | Cover.↑ | Divers.↑ | Frag.↑ | FCD↓ | Synth.↓ | $O_2$ Perm.↓ | $N_2$ Perm.↓ | $CO_2$ Perm.↓ | MAE↓ |
| *Molecule Optimization* | | | | | | | | | | |
| GraphGA | 100.00 (N/A) | 11/11 | 88.28 | 0.927 | 9.19 | 1.3307 | 1.9840 | 2.2900 | 1.9489 | 1.888 |
| MARS | 100.00 (N/A) | 11/11 | 83.75 | 0.928 | 7.56 | 1.1658 | 1.5761 | 1.8327 | 1.6074 | 1.546 |
| LSTM-HC | 99.10 (N/A) | 10/11 | 89.18 | 0.794 | 18.16 | 1.4251 | 1.1003 | 1.2365 | 1.0772 | 1.210 |
| JTVAE-BO | 100.00 (N/A) | 10/11 | 73.66 | 0.729 | 23.59 | **1.0714** | 1.0781 | 1.2352 | 1.0978 | 1.121 |
| *Continuous diffusion* | | | | | | | | | | |
| GDSS | 92.05 (90.76) | 9/11 | 75.10 | 0.000 | 34.26 | 1.3701 | 1.0271 | 1.0820 | 1.0683 | 1.137 |
| MOOD | 98.66 (92.05) | 11/11 | 83.49 | 0.023 | 39.40 | 1.4019 | 1.4961 | 1.7603 | 1.4748 | 1.533 |
| *Discrete diffusion* | | | | | | | | | | |
| DiGress v2 | 98.12 (30.57) | 11/11 | **91.05** | 0.278 | 21.73 | 2.7507 | 1.7130 | 2.0632 | 1.6648 | 2.048 |
| GraphDiT | 82.45 (84.37) | 11/11 | 87.12 | 0.960 | 6.64 | 1.2973 | 0.7440 | 0.8857 | 0.7550 | 0.921 |
| *Masked diffusion* | | | | | | | | | | |
| MDM w/ cosine | 15.95 (37.16) | 11/11 | 89.91 | 0.307 | 26.45 | 2.1795 | 1.5035 | 1.7755 | 1.4974 | 1.739 |
| MDM w/ polynomial | 18.61 (60.32) | 11/11 | 88.44 | 0.237 | 29.32 | 2.0041 | 1.6805 | 1.9846 | 1.6468 | 1.829 |
| MDM w/ power-law | 17.31 (53.64) | 11/11 | 89.08 | 0.401 | 26.56 | 2.0145 | 1.4100 | 1.6536 | 1.4030 | 1.620 |
| **MELD (Ours)** | 99.10 (96.51) | 11/11 | 85.91 | **0.974** | **5.93** | 1.1398 | **0.6433** | **0.7596** | **0.6496** | **0.798** |

## 5.2 MAIN RESULTS

**Unconditional Generation.** We present the results of **MELD** on QM9 and ZINC250K datasets for unconditional generation. Remarkably, **MELD** substantially enhances distributional similarity while maintaining perfect validity, as shown in Table 1. On the QM9 dataset, our method outperforms GraphARM (Kong et al., 2023), the autoregressive masked diffusion baseline, with up to 91% reduction in FCD and NSPDK. Moreover, it significantly improves the NSPDK by up to 98% from standard MDMs.

On the more challenging ZINC250K dataset, which includes larger molecules and richer atom types, **MELD** achieves state-of-the-art results on 5 out of 6 metrics, surpassing GruM (Jo et al., 2024), the strongest baseline. It also consistently improves over masked diffusion baselines on key metrics including FCD, NSPDK, and scaffold similarity (Scaf.). In contrast, standard MDMs exhibit degenerate behavior, with FCD 91.4% higher and a Scaf. 99.8% lower than the best diffusion-based baselines, suggesting that element-agnostic schedulers yield valid but distributionally misaligned molecules.

**Property-conditioned Generation.** Next, we evaluate **MELD** on conditional generation using the Polymer dataset (Thornton et al., 2012), with results summarized in Table 2. Overall, **MELD** establishes a new state-of-the-art in property alignment, with a 13.4% reduction in average MAE relative to GraphDiT (Liu et al., 2024a). Apart from GraphDiT, no existing method can satisfy multiple property constraints simultaneously: LSTM-HC achieves strong synthesizability MAE but

Table 3: Ablation study of **MELD** with varying noise scheduling approaches. $\gamma$ without $\phi$ and $\gamma_\phi$ denote fixed and learnable schedules, respectively. V.U.N. denotes a composite score for Validity, Uniqueness, and Novelty.

| Schedule type | Method | FCD↓ | NSPDK↓ | Scaf.↑ | V.U.N.↑ |
|---|---|---|---|---|---|
| Fixed $\gamma$ | Power-law | 26.09 | 0.0683 | 0.001 | **100.00** |
| | DiffusionBERT (He et al., 2022) | 1.95 | 0.0009 | 0.491 | **100.00** |
| Learnable $\gamma_\phi$ | GenMD4 (Shi et al., 2024) | 3.19 | 0.0017 | 0.429 | **100.00** |
| | TabDiff (Shi et al., 2025) | 2.15 | 0.0009 | 0.486 | 99.99 |
| | **MELD (Ours; Node)** | 1.63 | 0.0009 | 0.536 | 99.99 |
| | **MELD (Ours; Edge)** | 1.73 | 0.0009 | 0.525 | 99.99 |
| | **MELD (Ours; Node + Edge)** | **1.51** | **0.0006** | **0.559** | 99.96 |

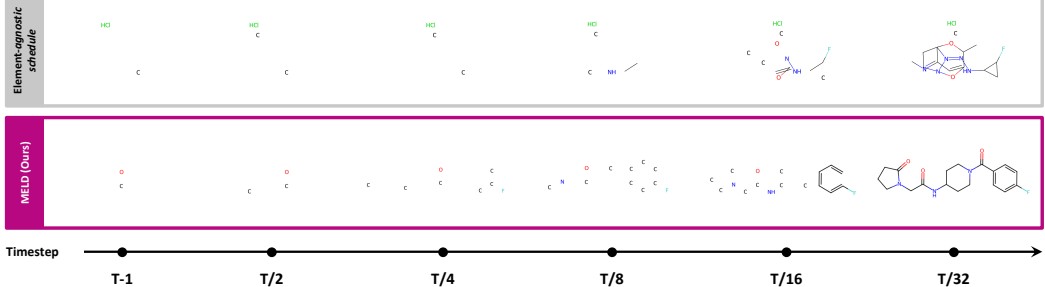

Figure 3: Comparison between fixed power-law scheduling and **MELD** during reconstruction. With the learnable noise schedule, **MELD** achieves faster recovery than standard MDMs.

fails under gas permeability targets. DiGress v2 (Vignac et al., 2023), despite leveraging classifier guidance (Dhariwal & Nichol, 2021), incurs substantially higher MAE across most conditions. Beyond alignment, **MELD** also improves generative quality, surpassing FCD and fragment-based similarity (Frag.) over the previous best. Consistent with earlier work (Ho & Salimans, 2021; Liu et al., 2024b), we observe an inherent trade-off between property alignment and sample diversity. Importantly, our method addresses the state-clashing issue prevalent in MDMs: whereas element-agnostic schedule results in generating low-quality molecules, our learnable, element-wise noise schedule enhances validity by a factor of five and improves property alignment by an average of 50%.

## 5.3 ABLATION STUDY

We evaluate several learnable scheduling strategies on ZINC250K (Irwin et al., 2012), as summarized in Table 3. The first row reports a standard MDM with a power-law function, while the second-to-last and third-to-last rows correspond to element-wise learnable scheduling applied only to nodes or edges. Rows two through four present advanced scheduling strategies from prior work. DiffusionBERT (He et al., 2022) employs a fixed spindle noise schedule decided by class-wise entropy; GenMD4 (Shi et al., 2024) is another class-wise scheduling variant where each atom and bond type has its own learned corruption rate; and TabDiff (Shi et al., 2025) introduces a single corruption rate shared across elements within the same column, analogous to node and edge-level schedules, *e.g.*, all nodes sharing the same schedule. The final row corresponds to the full element-wise schedule of **MELD**.

As depicted in the table, all alternative noise schedules fall short of optimal gains in key metrics such as FCD and Scaf., an effect we attribute to their limited ability of reducing state-clashing. For instance, employing GenMD4 noise scheduling can remain limited in resolving the state-clashing: delaying the corruption of all carbon atoms relative to nitrogen in o-Phenylenediamine (Figure 2) may still result in symmetric benzene ring. In contrast, our full per-element corruption (**MELD**) delivers further reductions in distributional similarity metrics, demonstrating its fine-grained control.

## 5.4 QUALITATIVE ANALYSIS

**Reverse process of MELD.** Figure 3 compares **MELD** with standard MDM. Corrupted nodes and edges are shown as [mask] and dashed lines, respectively. Under unified noise scheduling,

Table 4: Computational cost analysis with varying molecular sizes (batch size = 1). All experiments were conducted on an NVIDIA GeForce RTX 4090 GPU and an AMD EPYC 7K62 48-Core Processor. Runtime values are averaged over 5 random seeds.

| #Atoms | Method | #Params (M) | FLOPs (GMac) | Peak memory (MB) | Runtime (sec) |
|---|---|---|---|---|---|
| 10 | MDM | 156.23 | 2.06 | 622.76 | 0.056 ± 0.005 |
| | **MELD** | 156.24 | 2.06 | 623.19 | 0.056 ± 0.006 |
| 50 | MDM | 157.22 | 9.88 | 680.40 | 0.077 ± 0.007 |
| | **MELD** | 157.23 | 9.93 | 692.31 | 0.087 ± 0.008 |
| 100 | MDM | 158.46 | 19.74 | 755.12 | 0.093 ± 0.008 |
| | **MELD** | 158.47 | 19.94 | 788.71 | 0.098 ± 0.008 |
| 200 | MDM | 160.93 | 39.77 | 929.09 | 0.132 ± 0.007 |
| | **MELD** | 160.94 | 40.59 | 1067.64 | 0.165 ± 0.008 |

unmasking proceeds relatively slowly: at time $t = T/4$ only HCl and two carbon atoms begin to emerge, ultimately leading to a poorly-designed molecule. In contrast, **MELD** reconstructs fragments earlier relative to element-agnostic schedule, where larger amount of atoms already reveal from step $t = T/4$. Similar phenomena can also be found with more examples in Section D.7.

**Computational cost analysis**   In practice, the computational and memory overhead introduced by **MELD** is negligible since it only adds learnable embedding matrix $H$ to the existing transformer-based architectures. To validate this, we report (1) total number of parameters, (2) FLOPs, (3) peak memory usage, and (4) single-step runtime across various molecular sizes ($|\mathcal{V}| \in [10, 50, 100, 200]$) for **MELD** and standard MDM in Table 4. Our results demonstrate that **MELD** introduces only about 0.01M additional parameters with negligible computational overhead, regardless of the input size.

**Scalability to large molecules.**   We further evaluate **MELD** on the large-scale Guacamol dataset (Brown et al., 2019) following the standard protocol used in prior work (Vignac et al., 2023). As demonstrated in Table 5, **MELD** surpasses all diffusion-based baselines (Vignac et al., 2023; Xu et al., 2024) while achieving 100% validity. Notably, this performance

Table 5: Performance comparison on large-scale Guacamol dataset. The metrics are transformed such that higher values indicate better performance.

| Method | Valid.↑ | Uniq.↑ | Novel.↑ | KL div.↑ | FCD↑ |
|---|---|---|---|---|---|
| ConGress (Vignac et al., 2023) | 0.1 | **100.0** | **100.0** | 36.1 | 0.0 |
| DiGress (Vignac et al., 2023) | 85.2 | **100.0** | 99.9 | 92.9 | 68.0 |
| DisCo (Xu et al., 2024) | 86.6 | **100.0** | 99.9 | 92.6 | 59.7 |
| **MELD** | **100.0** | **100.0** | **100.0** | **93.4** | **68.8** |

is obtained with 70% reduced training epochs (300 epochs) than DiGress (1000 epochs), emphasizing both efficiency and empirical gains.

**Quantifying state-clashing problem.** Here, we assess state-clashing phenomenon by measuring the number of distinct intermediate graph states at each timestep, as shown in Table 6. Specifically, we sample molecules with a fixed graph size and employ a graph isomorphism-based method (Cordella et al., 2001) to count unique graphs. A

Table 6: Number of unique graph states across varying timesteps in ZINC250K, averaged over 3 seeds.

| Method | T-100 | T-75 | T-50 | T-25 | T-1 |
|---|---|---|---|---|---|
| MDM w/ cosine | **131.0** | 122.3 | 63.0 | 14.7 | 1.7 |
| MDM w/ polynomial | **131.0** | **131.0** | **131.0** | 103.0 | 13.3 |
| MDM w/ power-law | **131.0** | **131.0** | **131.0** | 126.0 | 8.7 |
| **MELD** | **131.0** | **131.0** | **131.0** | **131.0** | **17.3** |

higher count of unique graphs indicates fewer state-clashing. Due to the extreme cost of isomorphism algorithm, we sample 131 molecules with 12 nodes from the ZINC250K dataset for the evaluation. The results show that **MELD** preserves greater structural diversity at later timesteps compared to any standard MDMs.

It is important to note that **MELD** is not intended to eliminate state-clashing *entirely*, but to reduce the chance of its occurrence, particularly in the early and intermediate timesteps. Inevitably, some clashes remain, *e.g.*, all graphs converge to a fully masked state, but these unavoidable cases only affect a small portion of decisions near the prior distribution and therefore does not compromise its overall effectiveness.

**Generalizability on synthetic graph.** To assess generalizability of **MELD** on other discrete graph domains, we benchmark **MELD** against two strong molecular diffusion models, DiGress and GruM, on SBM (Martinkus et al., 2022), a synthetic graph benchmark. Following the standard evaluation protocol (Vignac et al., 2023; Jo et al., 2024), we compute the maximum mean discrepancy (MMD) across four key graph statistics.

Table 7: Performance comparison on synthetic graph domain (SBM).

| Method | Degree↓ | Cluster↓ | Orbit↓ | Spectral↓ | V.U.N.↑ |
|---|---|---|---|---|---|
| DiGress | 0.0013 | 0.0498 | 0.0434 | 0.0400 | 74.00 |
| GruM | 0.0007 | **0.0492** | 0.0448 | 0.0050 | 85.00 |
| **MELD** | **0.0005** | 0.0506 | **0.0381** | **0.0047** | **97.50** |

As reported in Table 7, **MELD** outperforms the baselines on most metrics, with notable gains in V.U.N. (a composite score for validity/uniqueness/novelty) and Orbit.

## 6 CONCLUSION

In this work, we investigated masked diffusion models (MDMs) for molecular graph generation and identified a central limitation, which we term *state-clashing*. To address this, we introduced **MELD**, a masked diffusion model that learns element-wise forward trajectories through a parameterized noise scheduling. Extensive experiments show that **MELD** consistently outperforms standard MDMs and prior diffusion-based methods in both unconditional and property-conditioned molecular generation.

### ETHICS STATEMENT

From a broader perspective, **MELD** has a potential to accelerate molecular discovery and reduce the need for costly and time-intensive wet-lab experiments, thereby contributing to advancements in drug design and material science. However, as with any generative technology, there exists the risk of misuse, including the malicious design of toxic or harmful compounds. We advocate for the responsible deployment of such models for the safe integration into real-world workflows.

### REPRODUCIBILITY STATEMENT

We provide the source code and setup for our key experiments, with detailed configurations described in the appendix. The implementation has been carefully verified, and we empirically confirm the validity of the proposed method.

### ACKNOWLEDGMENTS

This work was partly supported by Institute for Information & communications Technology Planning & Evaluation(IITP) grant funded by the Korea government(MSIT) (RS-2019-II190075, Artificial Intelligence Graduate School Support Program(KAIST)), National Research Foundation of Korea(NRF) grant funded by the Ministry of Science and ICT(MSIT) (No. RS-2022-NR072184), National Research Foundation of Korea(NRF) grant funded by the Korea government(MSIT) (RS-2025-02216257), GRDC(Global Research Development Center) Cooperative Hub Program through the National Research Foundation of Korea(NRF) grant funded by the Ministry of Science and ICT(MSIT) (No. RS-2024-00436165), Institute of Information & Communications Technology Planning & Evaluation(IITP) grant funded by the Korea government(MSIT) (RS-2025-02304967, AI Star Fellowship(KAIST)), and the National Supercomputing Center with supercomputing resources including technical support (KSC-2024-CRE-0535). We also thank Polymerize for their valuable support in improving the manuscript.

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

SUPPLEMENTARY MATERIALS

## A  MORE RELATED WORK

**Molecule optimization.**   Optimization-based methods generate molecules by iteratively refining candidates assembled from a predefined vocabulary of fragments, aiming to align with desired property constraints. These approaches typically employ techniques such as genetic algorithms (Jensen, 2019), Bayesian optimization (Shahriari et al., 2015; Jin et al., 2018; Zhu et al., 2023), and goal-directed generation (Mollaysa et al., 2020; You et al., 2018a). Representative examples include (Jin et al., 2020; 2018; Xie et al., 2021; Fu et al., 2022), which utilize predefined subgraph motifs or scaffolds to ensure chemical validity during the generation process. These methods rely on diverse strategies including Markov sampling to sparse Gaussian processes and optimize molecules based on property-specific scoring functions. Goal-directed generation (Mollaysa et al., 2020; You et al., 2018a), in particular, often adopts reinforcement learning, where a generation policy is updated to maximize a property-driven reward function. Despite their strengths, existing optimization-based approaches remain limited in conditional generation settings. Specifically, they require a full re-optimization for each new property configuration when tasked with generating molecules that precisely match target properties, rather than simply increasing or decreasing property values. This results in a high training complexity and limits their scalability (Aung et al., 2024; Xia et al., 2024).

**Learnable noise scheduling.**   Several works have explored learnable corruption process to optimize the forward trajectories in images and text. In continuous-space diffusion models, Kingma et al. (2021) introduces a learnable scalar noise schedule as a function of time, enabling variance reduction in evidence lower bound (ELBO) estimation. Extending this, Sahoo et al. (2024b) proposes a multivariate, data-dependent noise schedule, showing that a non-scalar, adaptive diffusion process can further tighten the ELBO by aligning the forward process more closely with the true posterior. In discrete masked diffusion, Shi et al. (2024) generalizes the corruption process to allow class-dependent masking rates across tokens, prioritizing semantically important tokens during generation. Shi et al. (2025) adopts feature-wise noise schedule for tabular data, where a single corruption rate is shared across elements within the same column. Additionally, Schrödinger bridges-based approaches (Peluchetti, 2023; De Bortoli et al., 2021; Shi et al., 2023) formulate generative modeling as learning an expressive, path-wise forward process by solving entropy-regularized optimal transport problems over path spaces.

It is noteworthy that the design philosophy of **MELD** is built upon the state-clashing, a critical issue that has not been addressed in these work nor in the molecular diffusion literature (Vignac et al., 2023; Jo et al., 2022; 2024; Liu et al., 2024a; Lee et al., 2023). While employing the learnable forward process, our work departs from existing methods by introducing graph element-wise parameterization of the forward diffusion, specifically to avoid trajectory collisions between semantically distinct molecules. Moreover, we explicitly target and resolves the intermediate state degeneracy unique to discrete molecular graphs, while Schrödinger bridge-based approaches neither address structural collapse in discrete settings nor differentiate forward paths across individual graph elements.

## B  LIMITATIONS

While our element-wise noise scheduling significantly mitigates the state-clashing issue, it may not fully address the inherent multimodality when a large portion of molecules are masked at later diffusion steps. This is especially pronounced at later diffusion steps, where a majority of the graph elements are masked, making it challenging to distinguish them. Nevertheless, these unavoidable cases only affects a small portion of corruption near the prior distribution and therefore does not compromise the overall efficacy of our method.

## C  EXPERIMENTAL SETUP

**Implementation details.**   We follow the evaluation protocols and dataset splits adopted in prior works: for unconditional generation, we adopt the setup from Jo et al. (2024), and for property-conditioned tasks, we follow the procedure outlined in Liu et al. (2024a). We provide the detailed

Table 8: Dataset statistics.

| Dataset | #(Graphs) | #(Nodes) | #(Node types) | #(Edge types) |
|---------|-----------|----------|---------------|---------------|
| QM9 | 133,985 | $|\mathcal{V}| \leq 9$ | 4 | 3 |
| ZINC250K | 249,555 | $|\mathcal{V}| \leq 38$ | 9 | 3 |
| Polymers | 553 | $|\mathcal{V}| \leq 50$ | 11 | 3 |

statistics of each dataset in Table 8. During training for unconditional generation, we apply an exponential moving average (EMA) to the model parameters, consistent with the training framework in Jo et al. (2024). For conditional generation, we utilize the implementation strategies proposed in Peebles & Xie (2023); Liu et al. (2024a), including condition vector encoders and adaptive layer normalization (AdaLN). Across all experiments, we use a transformer-based denoising model (Peebles & Xie, 2023) with 6 layers, a hidden dimension of 1152, and 16 attention heads. The noise scheduling network is parameterized as a two-layered MLP with SiLU activation with hidden dimension set as 64. We train all models using the AdamW optimizer with no weight decay. We provide detailed training setups compared with representative baselines (Jo et al., 2024; Vignac et al., 2023; Liu et al., 2024a) in Table 9. The FLOPS comparison with these baselines is shown in Table 10. Models are implemented in PyTorch Paszke et al. (2019) with PyTorch Geometric Fey & Lenssen (2019). Experiments were conducted on machines equipped with NVIDIA RTX 3090 and 4090 GPUs (24 GB) and AMD EPYC 7543 32-Core CPUs (64 cores total). Note that **MELD** does not rely on increased GPU count or specialized accelerators compared to baselines. For reference, the results of Jo et al. (2024) were obtained using RTX 3090 and 2080 Ti, while Liu et al. (2024a) used an A6000 GPU. These configurations all fall within a similar class of commodity GPUs, and none of the compared methods (including **MELD**) benefits from substantially larger compute bugets or hardware advantages.

Table 9: Training details comparison with representative baselines.

| Datasets | Methods | Total epochs | Diffusion sampling step | Learning rate | Scheduler | Backbone |
|----------|---------|--------------|-------------------------|---------------|-----------|----------|
| QM9 | DiGress | 1000 | 1000 | 2e−4 | Cosine | Graph transformer |
| | GruM | 1000 | 1000 | 2e−4 | Linear | Graph transformer |
| | **MELD** | 1500 | 300 | 2e−4 | Element-wise | Diffusion transformer (DiT) |
| ZINC250K | DiGress | 500 | 1000 | 2e−4 | Cosine | Graph transformer |
| | GruM | 500 | 1000 | 2e−4 | Linear | Graph transformer |
| | **MELD** | 600 | 500 | 2e−4 | Element-wise | Diffusion transformer (DiT) |
| Polymers | GraphDiT | 10000 | 500 | 2e−4 | Cosine | Diffusion transformer (DiT) |
| | **MELD** | 15000 | 200 | 2e−4 | Element-wise | Diffusion transformer (DiT) |

**Baselines.** We consider various recent baselines for conditional and unconditional generation; following experimental setups of prior works (Liu et al., 2024a; Jo et al., 2022; 2024).

- **Unconditional Generation**: First, we consider three flow-based models as baselines: MoFlow (Zang & Wang, 2020), GraphAF (Shi et al., 2020) and GraphDF (Luo et al., 2021), three continuous diffusion models: EDP-GNN (Niu et al., 2020), GDSS (Jo et al., 2022), and GruM (Jo et al., 2024), and one discrete-diffusion model: DiGress (Vignac et al., 2023). Additionally, we compare **MELD** against GraphARM (Kong et al., 2023), a method that employs mask tokens as absorbing states but generates tokens (*i.e.*, nodes) autoregressively.

- **Conditional Generation**: We consider four optimization-based frameworks as baselines: GraphGA (Jensen, 2019), MARS (Xie et al., 2021), LSTM-HC (Neil et al., 2018), and JTVAE-BO (Jin et al., 2018), two continuous diffusion models: GDSS (Jo et al., 2022) and MOOD (Lee et al., 2023), and two discrete diffusion models: DiGress v2 (Vignac et al., 2023) integrated with classifier guidance and GraphDiT (Liu et al., 2024a).

**Metrics.** Following the evaluation protocol in previous work (Liu et al., 2024a; Jo et al., 2022; 2024), we evaluate the performance of our framework using the following metrics:

- **Unconditional Generation**: We use 10,000 generated samples for evaluation using the following six metrics: (1) *Valid.*, the proportion of chemically valid molecules; (2) *Frechet ChemNet Distance* (FCD; Preuer et al. 2018), a distributional similarity score of ChemNet embeddings between generated and reference molecules; (3) *NSPDK* (Costa & De Grave, 2010), a graph

Table 10: Computational cost analysis comparing various methods (GruM, DiT, DiGress, and **MELD**). We report the average and standard deviation values of processing molecule size of $|V| = 100$, with a batch size of 32, upon 5 forward passes. All experiments were conducted on an NVIDIA GeForce RTX 4090 GPU and AMD EPYC 7K62 48-Core Processor.

| | Baselines | | | Masked Diffusion Models | | | |
|---|---|---|---|---|---|---|---|
| Metric | GruM | DiT | DiGress | Fixed Poly. | Fixed Cosine | TabDiff | **MELD** |
| FLOPs | $209.9 \pm 0.1$ | $314.7 \pm 0.1$ | $209.4 \pm 0.1$ | $315.2 \pm 0.1$ | $315.2 \pm 0.1$ | $318.5 \pm 0.1$ | $318.5 \pm 0.1$ |
| Exec (s) | $1.12 \pm 0.03$ | $0.91 \pm 0.01$ | $1.29 \pm 0.04$ | $0.94 \pm 0.02$ | $0.94 \pm 0.02$ | $0.95 \pm 0.02$ | $1.16 \pm 0.03$ |
| Peak (GB) | $19.16 \pm 0.05$ | $2.74 \pm 0.01$ | $19.15 \pm 0.06$ | $2.74 \pm 0.01$ | $2.74 \pm 0.01$ | $3.27 \pm 0.02$ | $2.77 \pm 0.01$ |

kernel metric that quantifies topological similarity to the reference set; (4) *Scaf.*, a scaffold-level similarity score; (5) *Uniqueness*, the proportion of valid molecules that are structurally distinct within the generated set; and (6) *Novelty*, the fraction of valid molecules not in the training data.

- **Conditional Generation**: We generate 10,000 samples and assess their overall quality using the following criteria: (1) *Valid.*, (2) *Cover.*, the heavy atom type coverage; (3) *Divers.*, the diversity among the generated molecules; (4) *Frag.*, a fragment-based similarity metric; and (5) FCD. We also report *Property Alignment*, measured as the mean absolute error (MAE) between target properties and the corresponding oracle-evaluated scores of generated molecules.

To compute property alignment, we follow the setup of prior works (Liu et al., 2024a; Gao et al., 2022), employing a random forest model trained on molecular fingerprints as an oracle function.

We report official baseline results (except for Kong et al. (2023)) from Jo et al. (2024) for unconditional generation and Liu et al. (2024a) for property-conditioned generation. For Kong et al. (2023), we take the results from the original paper. To evaluate the efficacy of our method in remedying state-clashing, we perform additional ablative studies against fixed-scheduling mechanisms often adopted in masked diffusion models; namely cosine (MDM w/ cosine), polynomial (MDM w/ polynomial), and power-law (MDM w/ power-law) scheduling functions. We train all vanilla MDM variants as well as **MELD** under identical training budgets. The corresponding standard deviation results are reported in Table 11 and Table 12. Note that the standard deviations for GraphARM and the conditional-generation baselines are not available.

# D FURTHER EXPERIMENTS AND ANALYSIS

## D.1 COMPARISON WITH MORE BASELINES

We present performance comparison between **MELD** and additional baselines Huang et al. (2023a;b). Note that the reproduced baseline performance differs from those in the original papers due to differences in formal charge assignment and bond order handling within the graph-to-molecule conversion pipeline, which we adopt from Jo et al. (2022; 2024). As shown in Table 13, **MELD** performs better or similar to CDGS for most of the metrics except for Wasserstein distance for logP scores. The performance is comparable to JODO, with better scores for FCD, WD/logP, WD/QED, WD/SA, worse scores for NSPDK, Scaf, and similar scores for Valid. Uniq. and Novel. Importantly, MELD attains this performance with significantly fewer total training iterations ( 7.6× reduction compared to CDGS; 5.7× compared to JODO) and with half the diffusion sampling steps, underscoring the efficiency of our method.

## D.2 ANALYSIS OF ELEMENT-WISE LEARNED EMBEDDING

Our design philosophy of learnable embedding in **MELD** is focused on reducing the chance of state-clashing problem by making each graph elements distinct and unique. As a result, our method can distinguish graph elements even within the symmetric motifs, which is often difficult to be discriminated using existing graph positional encodings (Dwivedi et al., 2022; Ma et al., 2023).

Table 14: Average cosine similarity between pairs of $\boldsymbol{h}^i$ and $\boldsymbol{h}^{ij}$ in a benzene ring.

| Cosine similarity | Learned $\boldsymbol{H}$ (**MELD**) | Random walk embedding |
|---|---|---|
| Nodes ($\boldsymbol{h}^i$) | 0.103 | 1 |
| Edges ($\boldsymbol{h}^{ij}$) | 0.237 | 1 |
| All | 0.192 | 1 |

Table 11: Unconditional generation performance of **MELD** with standard deviation. The baseline results are taken from Jo et al. (2024).

| Method | QM9 | | | | | |
| --- | --- | --- | --- | --- | --- | --- |
| | Valid.↑ | FCD↓ | NSPDK↓ | Scaf.↑ | Uniq.↑ | Novel.↑ |
| *Flow-based* | | | | | | |
| MoFlow | $91.36 \pm 1.23$ | $4.47 \pm 0.595$ | $0.017 \pm 0.003$ | $0.145 \pm 0.052$ | $\underline{98.65} \pm 0.57$ | $\underline{94.72} \pm 0.77$ |
| GraphAF | $74.43 \pm 2.55$ | $5.63 \pm 0.259$ | $0.021 \pm 0.003$ | $0.305 \pm 0.056$ | $88.64 \pm 2.37$ | $86.59 \pm 1.95$ |
| GraphDF | $93.88 \pm 4.76$ | $10.93 \pm 0.038$ | $0.064 \pm 0.000$ | $0.098 \pm 0.106$ | $98.58 \pm 0.25$ | $\mathbf{98.54} \pm 0.48$ |
| *Continuous diffusion* | | | | | | |
| EDP-GNN | $47.52 \pm 3.60$ | $2.68 \pm 0.221$ | $0.005 \pm 0.001$ | $0.327 \pm 0.115$ | $\mathbf{99.25} \pm 0.05$ | $86.58 \pm 1.85$ |
| GDSS | $95.72 \pm 1.94$ | $2.90 \pm 0.282$ | $0.003 \pm 0.000$ | $0.698 \pm 0.020$ | $98.46 \pm 0.61$ | $86.27 \pm 2.29$ |
| GruM | $\underline{99.69} \pm 0.19$ | $0.11 \pm 0.006$ | $\mathbf{0.0002} \pm 0.000$ | $\underline{0.945} \pm 0.005$ | $96.90 \pm 0.15$ | $24.15 \pm 0.80$ |
| *Discrete diffusion* | | | | | | |
| DiGress | $98.19 \pm 0.23$ | $\underline{0.10} \pm 0.008$ | $\underline{0.0003} \pm 0.000$ | $0.936 \pm 0.003$ | $96.67 \pm 0.24$ | $25.58 \pm 2.36$ |
| *Masked diffusion* | | | | | | |
| MDM w/ cosine | $\mathbf{100.00} \pm 0.00$ | $3.67 \pm 0.081$ | $0.009 \pm 0.001$ | $0.653 \pm 0.007$ | $85.96 \pm 0.62$ | $69.85 \pm 0.41$ |
| MDM w/ polynomial | $\mathbf{100.00} \pm 0.00$ | $3.70 \pm 0.093$ | $0.010 \pm 0.000$ | $0.890 \pm 0.006$ | $86.57 \pm 0.55$ | $67.18 \pm 0.38$ |
| MDM w/ power-law | $\mathbf{100.00} \pm 0.00$ | $3.62 \pm 0.074$ | $0.007 \pm 0.000$ | $0.628 \pm 0.006$ | $91.30 \pm 0.54$ | $76.65 \pm 0.36$ |
| **MELD** | $\mathbf{100.00} \pm 0.00$ | $\mathbf{0.09} \pm 0.004$ | $\mathbf{0.0002} \pm 0.000$ | $\mathbf{0.947} \pm 0.004$ | $96.49 \pm 0.13$ | $33.55 \pm 0.04$ |
| | ZINC250K | | | | | |
| Method | Valid.↑ | FCD↓ | NSPDK↓ | Scaf.↑ | Uniq.↑ | Novel.↑ |
| *Flow-based* | | | | | | |
| MoFlow | $63.11 \pm 5.17$ | $20.93 \pm 0.184$ | $0.046 \pm 0.002$ | $0.013 \pm 0.005$ | $\underline{99.99} \pm 0.01$ | $\mathbf{100.00} \pm 0.00$ |
| GraphAF | $68.47 \pm 0.99$ | $16.02 \pm 0.451$ | $0.044 \pm 0.005$ | $0.067 \pm 0.016$ | $98.64 \pm 0.69$ | $\underline{99.99} \pm 0.01$ |
| GraphDF | $90.61 \pm 4.30$ | $33.55 \pm 0.150$ | $0.177 \pm 0.001$ | $0.000 \pm 0.000$ | $99.63 \pm 0.01$ | $\mathbf{100.00} \pm 0.00$ |
| *Continuous diffusion* | | | | | | |
| EDP-GNN | $82.97 \pm 2.73$ | $16.74 \pm 1.300$ | $0.049 \pm 0.006$ | $0.000 \pm 0.000$ | $99.79 \pm 0.08$ | $\mathbf{100.00} \pm 0.00$ |
| GDSS | $97.01 \pm 0.77$ | $14.66 \pm 0.680$ | $0.019 \pm 0.001$ | $0.047 \pm 0.005$ | $99.64 \pm 0.13$ | $\mathbf{100.00} \pm 0.00$ |
| GruM | $\underline{98.65} \pm 0.25$ | $\underline{2.26} \pm 0.084$ | $\underline{0.0015} \pm 0.0003$ | $\underline{0.530} \pm 0.044$ | $99.97 \pm 0.03$ | $99.98 \pm 0.02$ |
| *Discrete diffusion* | | | | | | |
| DiGress | $94.99 \pm 2.55$ | $3.48 \pm 0.147$ | $0.0021 \pm 0.0004$ | $0.416 \pm 0.053$ | $99.97 \pm 0.01$ | $\underline{99.99} \pm 0.01$ |
| *Masked diffusion* | | | | | | |
| MDM w/ cosine | $\mathbf{100.00} \pm 0.00$ | $25.41 \pm 0.023$ | $0.051 \pm 0.0003$ | $0.001 \pm 0.000$ | $\underline{99.99} \pm 0.02$ | $\mathbf{100.00} \pm 0.00$ |
| MDM w/ polynomial | $\mathbf{100.00} \pm 0.00$ | $26.43 \pm 0.027$ | $0.053 \pm 0.0004$ | $0.001 \pm 0.000$ | $99.93 \pm 0.03$ | $\mathbf{100.00} \pm 0.00$ |
| MDM w/ power-law | $\mathbf{100.00} \pm 0.00$ | $26.09 \pm 0.031$ | $0.068 \pm 0.0004$ | $0.001 \pm 0.000$ | $\mathbf{100.00} \pm 0.00$ | $\mathbf{100.00} \pm 0.00$ |
| **MELD** | $\mathbf{100.00} \pm 0.00$ | $\mathbf{1.51} \pm 0.047$ | $\mathbf{0.0006} \pm 0.0001$ | $0.559 \pm 0.015$ | $\mathbf{100.00} \pm 0.01$ | $99.96 \pm 0.02$ |

Table 12: Property-conditioned generation performance of **MELD** with standard deviation, averaged over three runs. Note that the standard deviation of the baseline results, which are taken from Liu et al. (2024a), are unavailable.

| Method | General Quality | | | | Property Alignment | | | |
| --- | --- | --- | --- | --- | --- | --- | --- | --- |
| | Valid.↑ | Cover.↑ | Divers.↑ | Frag.↑ | FCD↓ | Synth.↓ | $O_2$ Perm.↓ | $N_2$ Perm.↓ | $CO_2$ Perm.↓ |
| MDM w/ cosine | $15.95 \pm 0.41$ $(37.16 \pm 0.28)$ | 11/11 | $\mathbf{89.91} \pm 0.12$ | $0.307 \pm 0.12$ | $\underline{26.45} \pm 0.180$ | $2.1795 \pm 0.014$ | $1.5035 \pm 0.021$ | $1.7755 \pm 0.030$ | $1.4974 \pm 0.019$ |
| MDM w/ polynomial | $\underline{18.61} \pm 0.52 (60.32 \pm 0.32)$ | 11/11 | $88.44 \pm 0.10$ | $0.237 \pm 0.005$ | $29.32 \pm 0.224$ | $\underline{2.0041} \pm 0.011$ | $1.6805 \pm 0.024$ | $1.9846 \pm 0.027$ | $1.6468 \pm 0.018$ |
| MDM w/ power-law | $17.31 \pm 0.52 (53.64 \pm 0.36)$ | 11/11 | $\underline{89.08} \pm 0.06$ | $\underline{0.401} \pm 0.009$ | $26.56 \pm 0.104$ | $2.0145 \pm 0.018$ | $\underline{1.4100} \pm 0.021$ | $\underline{1.6536} \pm 0.035$ | $\underline{1.4030} \pm 0.024$ |
| **MELD** | $\mathbf{99.10} \pm 0.12 (96.51 \pm 0.14)$ | 11/11 | $85.91 \pm 0.06$ | $\mathbf{0.974} \pm 0.001$ | $\mathbf{5.93} \pm 0.032$ | $\mathbf{1.1398} \pm 0.009$ | $\mathbf{0.6433} \pm 0.013$ | $\mathbf{0.7596} \pm 0.013$ | $\mathbf{0.6496} \pm 0.012$ |

To empirically verify this, we analyze the learned embedding matrix $\boldsymbol{H}$ on a benzene ring and compare it to random walk positional embeddings (Dwivedi et al., 2022). In Table 14, we evaluate the average pairwise cosine similarity for (1) node embeddings $\boldsymbol{h}^i$, (2) edge embeddings $\boldsymbol{h}^{ij}$ (connected to benzene ring), and (3) all element embeddings (nodes and edges). Our learned embeddings exhibit significantly low pair-wise similarity, suggesting that the learned embedding successfully distinguishes elements even within the symmetric structure.

## D.3 ROBUSTNESS ACROSS DIFFERENT DIFFUSION STEPS

We evaluate the performance of **MELD** on the Polymer dataset under varying diffusion steps, setting the total timestep $T \in \{50, 100, 150, 200\}$. Note that during this experiment, we fix the **MELD**-incorporated MDM to be trained upon a fixed diffusion step of 200, and only vary the number of steps taken during inference. We compare the performance of **MELD** against that of the strongest baseline, GraphDiT (Liu et al., 2024a), which is originally evaluated at diffusion step size of 500. As depicted in Figure 5, **MELD** overall exhibits robust performance across a range of metrics.

Table 13: Additional performance comparison of **MELD** with CDGS and JODO on ZINC250K dataset. WD represents Wasserstein distance between the distributions of molecular properties for generated and test molecules.

| Method | Total iter. | Diffusion steps | Valid.↑ | FCD↓ | NSPDK↓ | Scaf.↑ | Uniq.↑ | Novel.↑ | WD/logP↓ | WD/QED↓ | WD/SA↓ |
|---|---|---|---|---|---|---|---|---|---|---|---|
| CDGS | 2000000 | 1000 | **100.00** | 2.41 | 0.0007 | 0.475 | 99.97 | **99.99** | **0.220** | 0.008 | 0.292 |
| JODO | 1500000 | 1000 | **100.00** | 1.58 | **0.0004** | **0.599** | **100.00** | 99.94 | 0.464 | 0.022 | 0.380 |
| **MELD** | **263400** | **500** | **100.00** | **1.51** | 0.0008 | 0.560 | 99.99 | 99.95 | 0.340 | **0.004** | **0.110** |

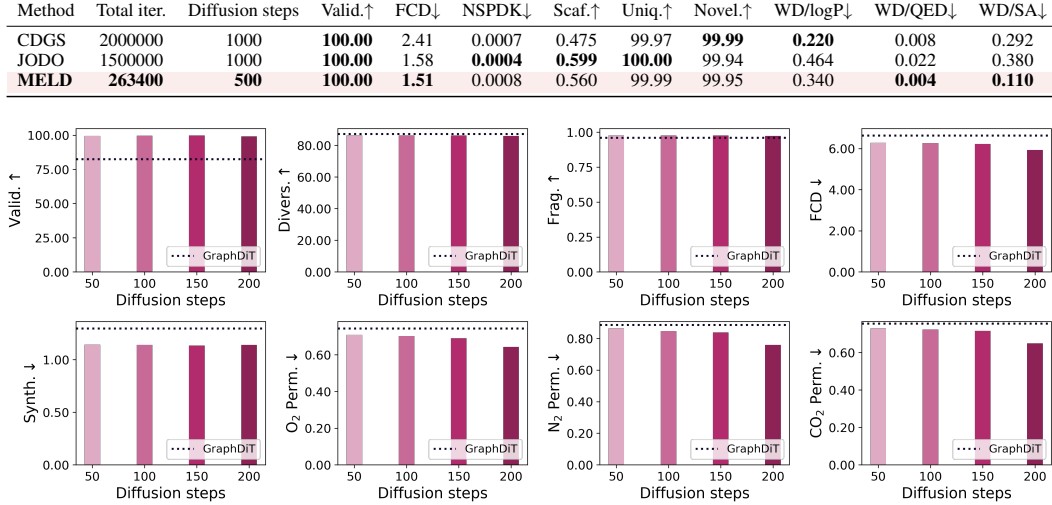

Figure 5: Performance of **MELD** under varying diffusion steps. The dotted line indicates the performance of the strongest baseline, GraphDiT, evaluated at a fixed diffusion step of 500.

### D.4 PER-ELEMENT SCHEDULING OF **MELD**

In Figure 4, we visualize the variation in per-step learned noise schedules across nodes and edges during the forward diffusion process. Specifically, we take 200 samples and plot the variation of the normalized masking probability $\sigma$, defined as the standard deviation of $\frac{\alpha_{t-1,\phi} - \alpha_{t,\phi}}{1 - \alpha_{t,\phi}}$.

We observe consistently higher variance for edge schedules across all timesteps, suggesting that the model prioritizes differentiating edges more aggressively than nodes during training. In addition, state-clashing problem is inherently intensified in the later steps of the forward process for both nodes and edges, as expected.

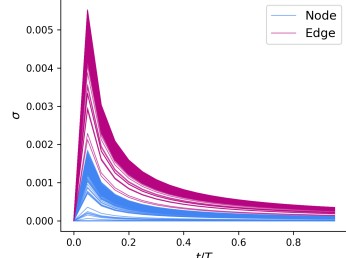

Figure 4: Variation of normalized masking probability $\sigma$.

### D.5 STATE-CLASHING ON SYNTHETIC GRAPH

While the state-clashing can also arise in other discrete graph domains such as citation graphs (Bernecker et al., 2024) or social networks (Ji et al., 2024), they typically involve a more diverse set of node and edge types, which reduces the likelihood that distinct graphs collapse into identical intermediate states. However, in synthetic graphs such as SBM (Martinkus et al., 2022), which contain only a single node type and binary edge types (denoting edge existence), state-clashing is susceptible to occur as depicted in Table 15.

Table 15: Number of unique graph states across varying timesteps in synthetic graph domain (SBM), averaged over 3 seeds.

| Method | T-100 | T-75 | T-50 | T-25 | T-1 |
|---|---|---|---|---|---|
| MDM w/ cosine | 36.7 | 23.3 | 11.7 | 4.3 | 1.3 |
| MDM w/ polynomial | **72.0** | **72.0** | 58.0 | 18.7 | **7.3** |
| MDM w/ power-law | **72.0** | 71.3 | 66.0 | 38.6 | 4.0 |
| **MELD** | **72.0** | **72.0** | **72.0** | **65.7** | 6.3 |

Using SBM synthetic graph (Martinkus et al., 2022) as a representative, we conducted the same quantitative analysis as done in Table 6. Due to the high computational cost of applying a full graph isomorphism check on the original graphs, we adopted a practical approximation: for each of 72 test and validation graphs, we randomly sampled 10 nodes and performed the state-clashing analysis.

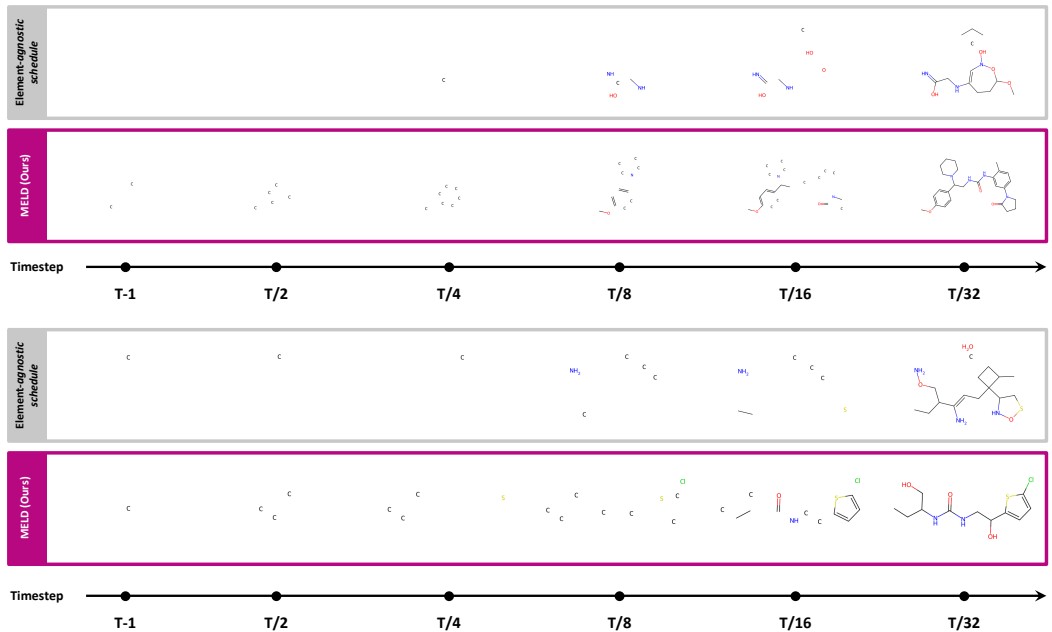

Figure 6: More comparisons between element-agnostic power-law scheduling and **MELD** during reconstruction on the ZINC250K dataset. With our proposed noise schedule, most reconstruction occurs at earlier timesteps relative to element-agnostic approach.

This procedure was repeated over 3 random seeds to ensure consistency. The results show that **MELD** mostly outperforms standard MDMs in terms of distinguishability.

### D.6 STATE-CLASHING ON LARGE-SCALE GRAPH

We further provide state-clashing analysis on the large-scale Guacamol dataset. Since direct graph isomorphism tests are computationally infeasible to quantify the degree of state-clashing, we use approximate fingerprint-based hashing method to count the number of graphs that are provably distinct. Specifically, for each intermediate graph, we compute a canonical graph fingerprint using a set of graph

Table 16: Number of graphs with distinct fingerprints across varying timesteps in the Guacamol dataset, averaged over 3 seeds.

| Method | T-100 | T-75 | T-50 | T-25 | T-1 |
|---|---|---|---|---|---|
| MDM w/ cosine | **144** | **144** | **144** | 134.3 | 4.7 |
| MDM w/ polynomial | **144** | **144** | **144** | **144** | 99.3 |
| MDM w/ power-law | **144** | **144** | **144** | **144** | 85.7 |
| **MELD** | **144** | **144** | **144** | **144** | **115.7** |

properties that remain unchanged under node relabeling. The fingerprint is constructed by combining the numbers of nodes and edges, sorted list of node degrees, and sorted multisets of node and edge types (including masked tokens). These components are concatenated into a canonical string (*e.g.*, n4|e3|d:1,1,2,2|v:0,6,6,8|eattr:1,1,2) and passed through a SHA-1 hashing. Graphs are counted as identical if their fingerprints match.

We sample molecules from Guacamol using the maximum graph size in its test set that has sufficient number of samples ($> 100$), resulting in collecting 144 molecules with 58 nodes each. The results in Table 16 below demonstrates that MELD retains substantially more distinct graph states at the last timestep than baselines, even for larger molecules.

### D.7 MORE EXAMPLES OF REVERSE PROCESS

We provide additional visualizations of reverse diffusion trajectories under **MELD** compared with those from a fixed power-law (element-agnostic) schedule in Figure 6. Consistent with our earlier analysis, **MELD** achieves faster reconstruction than standard MDMs. For instance, fragments begin

to unmask as early as $t = T - 1$, whereas the element-agnostic schedule only starts to recover them at $T/4 \leq t \leq T/8$.

### D.8 MOLECULE VISUALIZATION

In this section, we provide 2D visualization of molecules generated by **MELD**. As illustrated, **MELD** generates chemically realistic molecules even for polymers dataset with larger number of atoms (*i.e.*, $|\mathcal{V}| \leq 50$), verifying its robustness under various graph sizes.

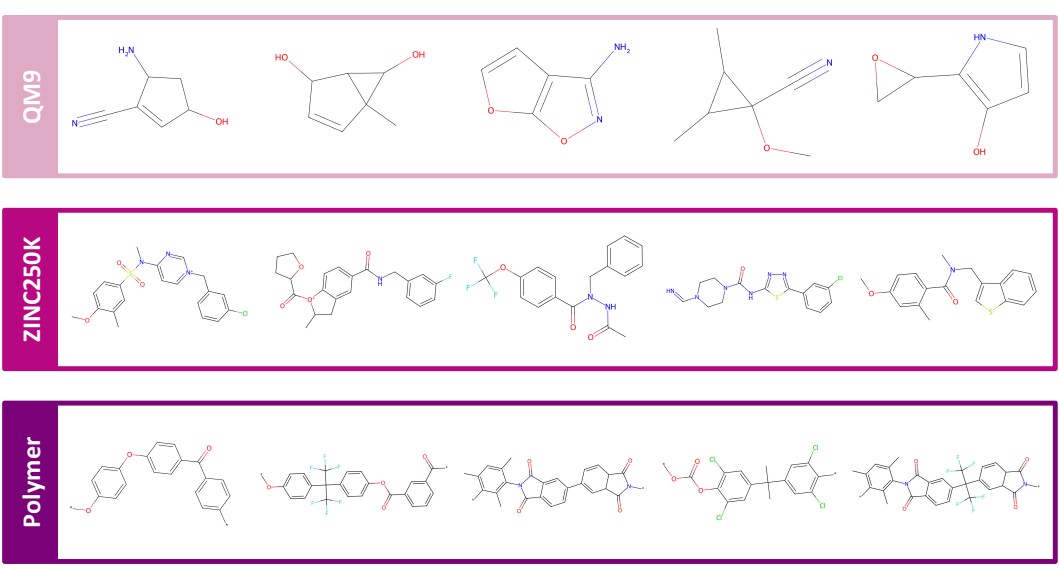

Figure 7: Visualization of molecules generated by **MELD**.

## E DISCUSSION WITH SUBSTITUTION-BASED CORRUPTION

In this section, we provide a more detailed discussion comparing our approach with substitution-based corruption methods. Indeed, substitution-based corruption (illustrated in Figure 8 (b)) can theoretically produce multimodal target distribution as well as masking-based corruption (depicted in Figure 8 (a)). However, the probability of creating multimodal targets is negligible in substitution-based methods compared to masking-based corruption. Below, we formalize this claim and demonstrate that state-clashing is a dominant failure mode for MDMs using a toy example.

**Formalizing state-clashing with collision probability.** Let $g$ denote a clean graph and $\tilde{g}_t$ the noisy graph at time $t$. Consistent with our manuscript, we denote $q_t(\tilde{g} \mid g)$ as the corruption kernel in the forward diffusion process. We define the state-clashing as the event where distinct clean graphs $g_1, g_2, ...$ collapse into the same noisy graph $\tilde{g}_t$ during the forward diffusion process. To quantify this, we consider the collision event $C_t$ at time $t$ for two distinct, independent graphs $g_1 \neq g_2$:

$$C_t(g_1, g_2) := \{\tilde{g}_1 = \tilde{g}_2 \mid g_1 \neq g_2\}, \quad \tilde{g}_1 \sim q_t(\cdot \mid g_1), \tilde{g}_2 \sim q_t(\cdot \mid g_2). \tag{8}$$

The severity of the collision is determined by the collision probability $\Pr[C_t(g_1, g_2)] = \sum_{\tilde{g}} q_t(\tilde{g} \mid g_1) q_t(\tilde{g} \mid g_2)$.

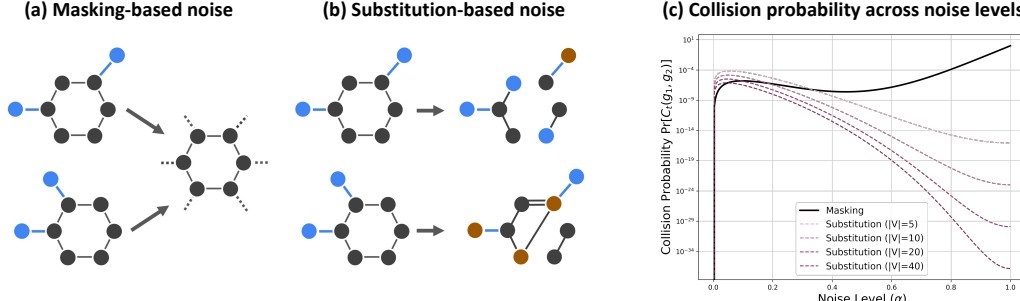

Figure 8: (a) Illustration of masking-based noising, and (b) illustration of substitution-based noising. (c) Quantitative comparison of state-clashing probabilities ($\Pr[C_t]$) between masked and substitution-based corruption. The analysis assumes $L = 23$ (average number of nodes in ZINC250K dataset) with $k = 2$ differing positions. While the collision probability for masking (solid black line) approaches 1 as the noise level $\alpha \to 1$, substitution-based methods (dashed lines) exhibit negligible probabilities that decrease further as vocabulary size $|\mathcal{V}|$ increases.

**Derivation of collision probability.** To compare two corruption approaches, consider a simplified scenario where graphs are sequences of nodes of length $L$ over a vocabulary $\mathcal{V}$. Each node is independently corrupted with probability $\alpha$ at specific timestep $t$. Take two clean sequences $\boldsymbol{g}_1$ and $\boldsymbol{g}_2$ that differ at exactly $k$ positions and are identical at $L - k$ positions. Note that while graphs are unordered, this element-wise comparison holds without loss of generality for permutation-equivariant denoisers.

(1) Case A: Differing positions ($\boldsymbol{g}_1(i) \neq \boldsymbol{g}_2(i)$)

- Masked corruption: we keep the original token with probability $1 - \alpha$, otherwise with probability $\alpha$ we replace it with a single special token $\texttt{[MASK]}$. A collision occurs only if both tokens are masked. Thus, the collision probability $p_{\text{diff}}^{\text{mask}}$ is computed as:

$$p_{\text{diff}}^{\text{mask}} = \Pr[\tilde{\boldsymbol{g}}_1(i) = \tilde{\boldsymbol{g}}_2(i) \mid \boldsymbol{g}_1(i) \neq \boldsymbol{g}_2(i)] = \alpha^2. \tag{9}$$

- Substitution corruption: we keep the original token with probability $1 - \alpha$, otherwise with probability $\alpha$ we randomly replace it with a token sampled uniformly from $\mathcal{V}$. A collision occurs if (i) one is kept and the other is substituted to that token, or (ii) both are substituted to the same token. This leads to the collision probability $p_{\text{diff}}^{\text{sub}}$ as follows:

$$p_{\text{diff}}^{\text{sub}} = \Pr[\tilde{\boldsymbol{g}}_1(i) = \tilde{\boldsymbol{g}}_2(i) \mid \boldsymbol{g}_1(i) \neq \boldsymbol{g}_2(i)] = 2 \cdot \frac{\alpha(1 - \alpha)}{|\mathcal{V}|} + \frac{\alpha^2}{|\mathcal{V}|} = \frac{\alpha(2 - \alpha)}{|\mathcal{V}|}. \tag{10}$$

(2) Case B: Identical positions ($\boldsymbol{g}_1(i) = \boldsymbol{g}_2(i)$)

- Masked corruption: A match occurs if both tokens are (i) masked or (ii) kept. Thus, the collision probability $p_{\text{eq}}^{\text{mask}}$ is computed as:

$$p_{\text{eq}}^{\text{mask}} = \alpha^2 + (1 - \alpha)^2. \tag{11}$$

- Substitution corruption: The corrupted tokens match if (i) both tokens are retained or (ii) both are substituted to the same token or (iii) one is kept and the other is substituted to the same token. The collision probability for $p_{\text{eq}}^{\text{sub}}$ is:

$$p_{\text{eq}}^{\text{sub}} = (1 - \alpha)^2 + \frac{\alpha^2}{|\mathcal{V}|} + 2\frac{\alpha(1 - \alpha)}{|\mathcal{V}|} = (1 - \alpha)^2 + \frac{\alpha(2 - \alpha)}{|\mathcal{V}|}. \tag{12}$$

Assuming independence across positions, the total collision probability is the product over all sites:

$$\Pr[C_t(\boldsymbol{g}_1, \boldsymbol{g}_2)]_{\text{mask}} = (p_{\text{diff}}^{\text{mask}})^k (p_{\text{eq}}^{\text{mask}})^{L-k}, \quad \Pr[C_t(\boldsymbol{g}_1, \boldsymbol{g}_2)]_{\text{sub}} = (p_{\text{diff}}^{\text{sub}})^k (p_{\text{eq}}^{\text{sub}})^{L-k}. \tag{13}$$

For positions where two graph elements differ, the ratio of collision probabilities is:

$$p_{\text{diff}}^{\text{mask}} \over p_{\text{diff}}^{\text{sub}} = \frac{\alpha^2}{\frac{\alpha(2-\alpha)}{|\mathcal{V}|}} = \frac{|\mathcal{V}|\alpha}{2-\alpha}. \tag{14}$$

This ratio exceeds 1 whenever $\alpha > \frac{2}{|\mathcal{V}|+1}$. For a typical vocabulary size in molecular datasets (*e.g.*, $|\mathcal{V}| = 9$ for ZINC250K) masked corruption yields a higher collision probability for all $\alpha > 0.2$. A similar trend applies to identical positions ($p_{\text{eq}}$). Since the term $(1-\alpha)^2$ is common to both methods, the inequality is determined by the remaining corruption terms, which follows the same ratio derived above.

Crucially, in the intermediate and later stages of diffusion (where $\alpha \to 1$), this gap becomes extreme. As $\alpha \to 1$, the collision probability for masking approaches 1, whereas for substitution, it approaches $(1/|\mathcal{V}|)^L$, where distinct graphs are highly scattered. That is, $\lim_{\alpha \to 1} \frac{\Pr[C_t(\boldsymbol{g}_1, \boldsymbol{g}_2)]_{\text{mask}}}{\Pr[C_t(\boldsymbol{g}_1, \boldsymbol{g}_2)]_{\text{sub}}} \approx |\mathcal{V}|^L$.

We visualize the gap between $\Pr[C_t(\boldsymbol{g}_1, \boldsymbol{g}_2)]_{\text{mask}}$ and $\Pr[C_t(\boldsymbol{g}_1, \boldsymbol{g}_2)]_{\text{sub}}$ in Figure 8 (c). The plot confirms that the gap becomes extreme as $\alpha \to 1$. Specifically, when $\alpha = 1$, the ratio $\Pr[C_t(\boldsymbol{g}_1, \boldsymbol{g}_2)]_{\text{mask}} / \Pr[C_t(\boldsymbol{g}_1, \boldsymbol{g}_2)]_{\text{sub}} \approx 10^{16}$. This confirms that while substitution can theoretically produce multimodal targets, the probability of such an event is orders of magnitude lower than masking with fair amount of vocabulary. Thus, state-clashing is much more pertinent to MDMs.

## F   COMPARISON BETWEEN STATE-CLASHING AND SYMMETRY-BREAKING

Several prior work (Lawrence et al., 2025; Laabid et al., 2025; Wang et al., 2024; Kaba & Ravanbakhsh, 2022) have introduced the concept of symmetry-breaking in self-symmetric inputs. For instance, Lawrence et al. (2025) and Laabid et al. (2025) identified that equivariant denoiser is unable to yield less symmetric output from highly self-symmetric noisy input. In this section, we provide two examples where state-clashing and symmetry-breaking formulations clearly differ.

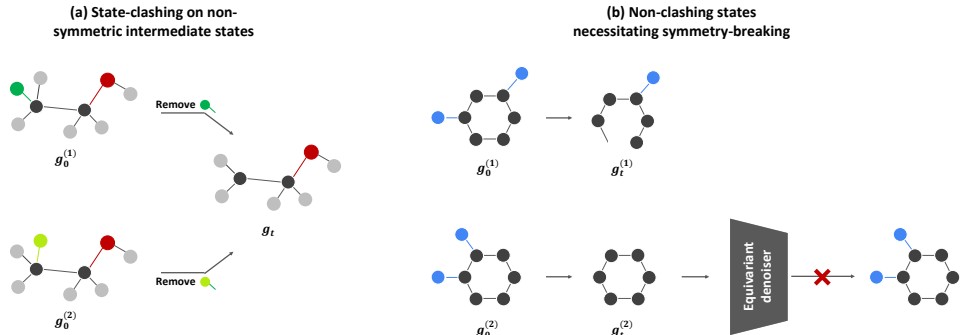

Figure 9: (a) Illustration of state-clashing between two non-isomers, collapsing into the asymmetric intermediate state. (b) Example case when the equivariant denoiser fails to reconstruct the original molecule from $\boldsymbol{g}_t^{(2)}$, although state-clashing does not happen.

We point out that state-clashing can occur even for non-symmetric cases, not only covering self-symmetric inputs. As illustrated in Figure 9 (a), masking the fluorine atom and bond in 2-Fluoroethanol ($\boldsymbol{g}_0^{(1)}$) produces the same masked state as masking the chlorine atom and bond in

Table 17: Comparison between **MELD** and SymPE on ZINC250K dataset.

| Method | Valid.↑ | FCD↓ | NSPDK↓ | Scaf.↑ | Uniq.↑ | Novel.↑ |
|---|---|---|---|---|---|---|
| SymPE | **100.00** | 8.24 | 0.0130 | 0.261 | 99.97 | **100.00** |
| **MELD** | **100.00** | **4.64** | **0.0044** | **0.370** | **99.99** | **100.00** |

2-Chloroethanol ($\boldsymbol{g}_0^{(2)}$). The collapsed masked state is not self-symmetric, yet the denoiser still faces a multimodal reconstruction target. Our emphasis is on mismatch between the joint distribution and the product of marginal distribution for denoising labels, *e.g.*, $p(x_1, x_2) \neq p(x_1)p(x_2)$. This can happen even when there exists no symmetry in intermediate states nor the reconstruction targets.

In contrast, symmetry breaking can help even when there exist no state-clashing, which we depict an example in Figure 9 (b). Here, even when no state-clashing happens between distinct graphs,

the reconstruction will fail for permutation-equivariant neural networks. Thus, state-clashing and symmetry breaking refers to distinct problems arising from lack of expressive power in (a) element-wise independent prediction and (b) permutation equivariant architectures used by the decoder, respectively. To compare our method with symmetry breaking approaches, we provide a direct empirical comparison against SymPE (Lawrence et al., 2025) on the ZINC250K dataset. Following the experimental setup in Lawrence et al. (2025), we adopt the graph transformer denoiser used in DiGress and evaluate both methods under the identical MDM framework. As demonstrated in Table 17, **MELD** consistently outperforms SymPE across all key metrics. In particular, MELD reduces FCD by nearly 50%.

