# OpenReview forum: "Learning Flexible Forward Trajectories for Masked Molecular Diffusion"
_ICLR.cc/2026/Conference — ICLR 2026 Poster_

### Official Review · Reviewer_88F6 · 2025-10-28

**Soundness:** 3
**Presentation:** 2
**Contribution:** 2
**Rating:** 2
**Confidence:** 3

**Summary:**

The authors identify state-calshing as a key limitation of Masked Diffusion Models (MDMs), which refers to the way graph components with different labels end up in the same intermediate states when using a unified forward process. This issue
makes denoising difficult, especially in the highly symmetrical space of molecular structures. To mitigate this issue,
the authors propose state-dependent learned forward processes to collect structural information about the molecule and maitain the partial identity of graph elements during denoising. The paper compares to relevant baselines in diffusion-based molecular generation and shows empirical improvements in some metrics.

**Strengths:**

- The paper identifies a specific limitation in MDMs and addresses it appropriately.
- The algorithm is clearly described.
- The main baselines in molecular generation are included in the empirical analysis.

**Weaknesses:**

**TL;DR** the weakest points for me are the positioning w.r.t the symmetry breaking literature, and the comparison to other noise methods used in diffusion (e.g. uniform, marginal, etc). I am willing to change my score if the author clarify these points and address some presentation issues.

**Connection to the symmetry breaking literature**: I understand that MELD seeks to prevent the state-clashing problem from happening in the first place, but how does it compare to methods using an equivariant forward process and denoiser, and breaking the symmetry of the 'highly similar' intermediate states during denoising? In other words, what is the connection between your work and the broader topic of symmetry breaking as tackled by [1,2,3,4]? I am asking in particular because MELD seems to sacrifice equivariance entirely, I am curious to see how it compares to models using the inductive bias of equivariance (e.g. to model the symmetries inherent to molecules and other graphs) while breaking the symmetry introduced by the forward process (highly similar masked intermediates).

**Claims regarding 'substitution-based corruption methods'**: The authors explain that other noising approaches (I am assuing methods like uniform or marginal noise) suffer from state clashing less than MDMs. While noising with random labels *looks* less similar than noising with the same mask token, uniformally generated intermediates should have the same amount of information as masked intermediates (i.e. in the o-Phenylenediamine and m-Phenylenediamine molecules, a uniform noise schedule should struggle as much as MDMs with recovering the right isomer). I find the claim that such methods 'preserve structural similarity and retain partial identity' unsubstantiated. Can you elaborate more on this?

**Presentation**:
- Since the state clashing problem is the main motivation of the paper, it would be more natural to present it first in the methods section and to emphasize it more in earlier sections (introductions, related work).
- Some claims in the paper are exagerated. For example, saying that applying diffusion to molecules is "underexplored" (abstract and introduction line 51), or that MELD achieves notable empirical improvements over the baselines.

**Questions:**

- How does MELD perform compared to equivariant models using symmetry breaking?
- Why do you think other corruption methods maintain partial identity through the forward process?

**Typos & nitpicks**
- Line 39: scalability
- Line 90: scalability and generalizability
- Line 255: '... as less probability of state-clashing,...' => incomplete phrase
- Line 457: '... a higher count of represents' => missing word

## References
[1] "Improved Equivariant Networks with Probabilistic Symmetry Breaking", Lawrence et al., 2025.

[2] "Equivariant Denoisers Cannot Copy Graphs: Aligns Your Graph Diffusion Models", Laabid et al., 2025.

[3] "Discovering symmetry breaking in physical systems with relaxed group convolution", Wang et al., 2024.

[4] "Equivariant networks for crystal structures, Kaba et al., 2022.

---

> ### Author Response · Authors · 2025-11-21
> **Official Comment by Authors (1)**
>
> We sincerely appreciate the reviewer’s feedback. We would like to address the reviewer’s concerns as below.
>
> ---
>
> **[W1 & Q1. Comparison with symmetry-breaking literature]**
>
> Thank you for your constructive comment and suggesting important references. To make our comparison more concrete, we describe each of the prior works one-by-one.
>
> [1,2] address the issue that arises when a standard equivariant denoiser is trained on targets that are marginalized over permutations, *i.e.*, state-clashing between randomly permuted targets. They resolve this by introducing symmetry breaking in inputs and constructing a canonical mapping between symmetry-broken input and one of the overlapping targets (via positional encoding or skip-connection), so that a consistent target can be chosen among multiple symmetric ones. In some sense, our state-clashing problem is generalization of the problems solved by symmetry breaking [1,2] applied to graph generation. These prior works solve the state-clashing caused by permutation of generation targets, and use symmetry breaking to construct a canonical mapping between symmetry-broken input and one of the overlapping targets. In contrast, our approach aims to alleviate arbitrary state-clashing issue by minimizing the amount of overlap, which helps avoid larger amount of cases like the isomer examples provided in Figure 1.
>
> [3] focuses on discovering symmetries required to solve supervised learning tasks in physical systems. To achieve this, [3] trains relaxed convolution layers, expressed as group element-weighted linear combination of filters, to identify which symmetries are broken in the data based on inspecting the learned group element-wise weights. In comparison, [3] focuses on supervised learning tasks while our work focuses on distribution learning. [3] aims to discover of unknown symmetries, while the permutation symmetry is known for our problem.
>
> [4] proposes an approach to systematically build message passing neural networks given symmetry of the underlying crstal system. It mentions that symmetry breaking can be necessary for describing phenomena spanning multiple unit cells and adds unit cell encodings as atom-wise positional encodings within a supercell. However, [4] does not further investigate how such symmetry-related issues arise during generative diffusion nor how to prevent trajectory-level collapse, which distinguishes it from our work.
>
> To further address your concern, we provide a direct empirical comparison between our method and SymPE [1] on the ZINC250K dataset. Following the experimental setup of [1], we adopt the graph transformer denoiser used in DiGress and evaluate both methods under the identical MDM framework. As shown in Table 1, MELD consistently outperforms SymPE across all key metrics. In particular, MELD reduces FCD by nearly 50%.
>
> Table 1. Comparison between MELD and SymPE on ZINC250K dataset.
> | Methods | Valid.    | FCD      | NSPDK      | Scaf.     | Uniq.      | Novel.    |
> | --------| --------- | -------- | ---------- | --------- | ---------- | --------- |
> | SymPE   | **100.0** | 8.24     | 0.0130     | 0.261     | 99.97      | **100.0** |
> | MELD    | **100.0** | **4.64** | **0.0044** | **0.370** | **99.99**  | **100.0** |

---

> > ### Comment · Reviewer_88F6 · 2025-11-27
> > **How is state-clashing a generalization of the problems identified by symmetry-breaking?**
> >
> > Thank you for providing additional results comparing to SymPE. This is indeed a helpful comparison, but I think your clarification regarding symmetry breaking vs state-clashing is inaccurate.
> >
> > [1] and [2] talk about the symmetry arising in the noisy space of masked graphs, so essentially the same as your setup, and not really 'the symmetry arising from randomly permuted graphs'. The issue raised by both works is that an equivariant denoiser would not be able to map between a highly self-symmetric noisy input (one where multiple states clash to the same labels) to a less self-symmetric output space (one representing a denoiser target, regardless of its permutation). For instance, the example you have in Figure 1 is covered by the symmetry problem identified by [1] and [2], both isomers would produce the same self-symmetric noisy graph, and an equivariant denoiser would learn a distribution proportional to the marginal distribution of labels in the dataset, and can generate the same output as shown in subfigure (a).
> >
> > I am still not sure then how state-clashing is different from high symmetry in the noisy space. Does state-clashing identify cases not covered by the framework of self-symmetry?

---

> ### Author Response · Authors · 2025-11-21
> **Official Comment by Authors (2)**
>
> **[W2 & Q2. Comparison with substitution-based corruption]**
>
> We thank the reviewer for their insightful feedback. From an information-theoretic perspective, a substitution-based noise should be just as challenging as noising with a [MASK] token to denoise, as both states are equally uninformative. However, this viewpoint assumes a denoiser with infinite expressive power to learn the true, highly multimodal posterior $p(g|g_{t})$; such an assumption does not hold in practice.
>
> To be specific, the denoising models $p_{\theta}(g_{t-1}|g_t)$ used in MDMs are a product of independent predictions for each node and edge, thus, unimodal. When two distinct molecules (o-Phenylenediamine and m-Phenylenediamine) are corrupted to an identical intermediate state $g_t$, it creates a multimodal target. This mismatch forces the model to learn a "sum" target, encouraging the model to spread its probability mass into an averaged state, often generating distributionally misaligned or invalid molecules.
>
> This brings back to the reviewers' challenge to our "preserve...partial identity". By utilizing absorbing states, *i.e.*, [MASK], the forward process *guarantees* multimodal targets through an identical intermediate state $g_t$. On the other hand,  substitution-based corruption "preserves more structural variation" because it uses the full vocabulary. While two different atoms could be substituted with the same atom (*e.g.*, N $\rightarrow$ O and C $\rightarrow$ O), it is far more likely that they are substituted with different atoms (*e.g.*, N $\rightarrow$ O and C $\rightarrow$ F), producing two distinct corrupted graphs, $g_{t,1}$ and $g_{t,2}$. This distinction is enough to prevent a state-clashing and keeps the target unimodal and learnable for our "product" model.
>
> We acknowledge that this explanation can be further clarified in our manuscript, and have revised it in accordance with the response (please refer to Appendix F in our updated manuscript).
>
> ---
>
> **[W3. Typos & Presentation]**
>
> Thank you for your helpful suggestions. We have corrected the typos and revised expressions that appeared exaggerated in the updated manuscript. In the revised version, we have introduced the state-clashing issue at the beginning of the method section as the reviewer suggested and emphasized in introduction and related work sections. Additionally, we would like to clarify that our statement refers specifically to masked diffusion models being underexplored in molecular generation, not standard diffusion models.
>
> **Reference**
>
> ---
>
> [1] Improving Equivariant Networks with Probabilistic Symmetry Breaking, ICLR '25.
>
> [2] Equivariant Denoisers Cannot Copy Graphs: Align Your Graph Diffusion Models, ICLR '25.
>
> [3] Discovering Symmetry Breaking in Physical Systems with Relaxed Group Convolution, ICML '24.
>
> [4] Equivariant Networks for Crystal Structures, NeurIPS '22.

---

> > ### Comment · Reviewer_88F6 · 2025-11-27
> > **Not fully convinced vis-a-vis substitution noise.**
> >
> > Thank you for providing a discussion in Appendix F. I am not fully convinced yet that a substitution-based noise schedule would not produce multi-modal targets. In the example you gave, even assuming the two isomers end up being noised to C->N and C->F, therefore producing different intermediate graphs, said graphs would **still look the same to a denoiser**. Unless the assumption here is that the denoiser could learn somehow that the N noisy token is more likely to produce isomer 1? I am not sure this assumption holds though if the noisy labels are uniformally sampled for each token... even if two isomers end up in different intermediate states, the two said states are unlikely to give the model any hints about the correct denoising output, so this is still technically the same 'approximating a multi-modal distribution with a uni-modal model' problem.
> >
> > Why do you think the 'uniform noise is as bad as mask noise' assumption requires a denoiser with infinite expressive power to learn the true, highly multimodal posterior $p(g|g_{t})$? I don't see how this is related to the denoiser's expressiveness. My point is that both types of noise lead to the same multi-modal target.

---

> > > ### Author Response · Authors · 2025-11-30
> > > **Further clarification on comparison with substitution-based corruption (1)**
> > >
> > > Thank you for your thoughtful follow-up. We appreciate the opportunity to clarify our points. Now we can further understand your point. We realize that our previous explanation was misleading and could be misinterpreted as substitution creates a mathematically strictly unimodal posterior, which is not what we intend to claim. We clarify that the probability of creating multimodal targets is negligible in substitution-based methods compared to masking-based corruption.
> > >
> > > Below, we formalize this claim and demonstrate that state-clashing is a dominant failure mode for MDMs via a toy example.
> > >
> > > **1. Formalizing state-clashing with collision probability**
> > >
> > > Let $g$ denote a clean graph and $\tilde g_t$ the noisy graph at time $t$. Consistent with our manuscript, we denote $q_t(\tilde g\mid g)$ as the corruption kernel in the forward diffusion process. We define the state-clashing as the event where distinct clean graphs $g_1,g_2,...$ collapse into the same noisy graph $\tilde g_t$ during the forward diffusion process. To quantify this, we consider the collision event $C_t$ at time $t$ for two distinct, independent graphs $g_1\ne g_2$:
> > >
> > > $$
> > > C_t(g_1,g_2):=\{\tilde g_1=\tilde g_2\mid g_1\ne g_2\},\quad \tilde g_1\sim q_t(\cdot\mid g_1), \tilde g_2\sim q_t(\cdot\mid g_2).
> > > $$
> > >
> > > The severity of the collision is determined by the collision probability $\mathrm{Pr}[C_t(g_1,g_2)]=\sum_{\tilde g} q_t(\tilde g\mid g_1)q_t(\tilde g\mid g_2)$.
> > >
> > > **2. Derivation of collision probability**
> > >
> > > To compare two corruption approaches, consider a simplified scenario where graphs are sequences of nodes of length $L$ over a vocabulary $\mathcal V$. Each node is independently corrupted with probability $\alpha$ at specific timestep $t$.
> > >
> > > Take two clean sequences $g_1$ and $g_2$ that differ at exactly $k$ positions and are identical at $L-k$ positions (While graphs are unordered, this element-wise comparison holds without loss of generality for permutation-equivariant denoisers).
> > >
> > > **Case A: Differing positions ($g_1(i)\ne g_2(i)$)**
> > >
> > > - Masked corruption: we keep the original token with probability $1-\alpha$, otherwise with probability $\alpha$ we replace it with a single special token [MASK]. A collision occurs only if both tokens are masked. Thus, the collision probability $p_{\mathrm{diff}}^{\mathrm{mask}}$ is computed as:
> > > $$
> > > p_{\mathrm{diff}}^{\mathrm{mask}}=\text{Pr}[\tilde g_1(i)=\tilde g_2(i)\mid g_1(i)\ne g_2(i)]=\alpha^2.
> > > $$
> > > - Substitution corruption: we keep the original token with probability $1-\alpha$, otherwise with probability $\alpha$ we randomly replace it with a token sampled uniformly from $\mathcal V$. A collision occurs if (i) one is kept and the other is substituted to that token, or (ii) both are substituted to the same token. This leads to the collision probability $p_{\mathrm{diff}}^{\mathrm{sub}}$ as follows:
> > > $$
> > > p_{\mathrm{diff}}^{\mathrm{sub}}=\mathrm{Pr}[\tilde g_1(i)=\tilde g_2(i)\mid g_1(i)\ne g_2(i)]=2\cdot\frac{\alpha(1-\alpha)}{|\mathcal V|} + \frac{\alpha^2}{|\mathcal V|}=\frac{\alpha(2-\alpha)}{|\mathcal V|}.
> > > $$
> > >
> > > **Case B: Identical positions ($g_1(i)= g_2(i)$)**
> > >
> > > - Masked corruption: A match occurs if both tokens are (i) masked or (ii) kept. Thus, the collision probability $p_{\mathrm{eq}}^{\mathrm{mask}}$ is computed as:
> > > $$
> > > p_{\mathrm{eq}}^{\mathrm{mask}}=\alpha^2 + (1-\alpha)^2.
> > > $$
> > > - Substitution corruption: The corrupted tokens match if (i) both tokens are retained or (ii) both are substituted to the same token or (iii) one is kept and the other is substituted to the same token. The collision probability for $p_{\mathrm{eq}}^{\mathrm{sub}}$ is:
> > > $$
> > > p_{\mathrm{eq}}^{\mathrm{sub}}=(1-\alpha)^2+\frac{\alpha^2}{|\mathcal V|} + 2\frac{\alpha(1-\alpha)}{|\mathcal V|}=(1-\alpha)^2+\frac{\alpha(2-\alpha)}{|\mathcal V|}.
> > > $$
> > >
> > > Assuming independence across positions, the total collision probability is the product over all sites:
> > > $$
> > > \Pr\left[C_t(g_1,g_2)\right]^{\mathrm{mask}}
> > > = \left(p_{\mathrm{diff}}^{\mathrm{mask}}\right)^k \left(p_{\mathrm{eq}}^{\mathrm{mask}}\right)^{L-k}, \quad \Pr\left[C_t(g_1,g_2)\right]^{\mathrm{sub}} = \left(p_{\mathrm{diff}}^{\mathrm{sub}}\right)^k \left(p_{\mathrm{eq}}^{\mathrm{sub}}\right)^{L-k}.
> > > $$
> > >
> > > For positions where two graph elements differ, the ratio of collision probabilities is:
> > >
> > > $$
> > > \frac{p_{\mathrm{diff}}^{\mathrm{mask}}}{p_{\mathrm{diff}}^{\mathrm{sub}}} = \frac{\alpha^2}{\frac{\alpha(2-\alpha)}{|\mathcal V|}}=\frac{|\mathcal V|\alpha}{2-\alpha}.
> > > $$
> > >
> > > This ratio exceeds 1 whenever $\alpha>\frac{2}{|\mathcal V|+1}$. For a typical vocabulary size in molecular datasets (*e.g.*, $|\mathcal V|=9$ for ZINC250K) masked corruption yields a higher collision probability for all $\alpha > 0.2$. A similar trend applies to identical positions ($p_{\text{eq}}$). Since the term $(1-\alpha)^2$ is common to both methods, the inequality is determined by the remaining corruption terms, which follows the same ratio derived above.

---

> > > > ### Author Response · Authors · 2025-11-30
> > > > **Further clarification on comparison with substitution-based corruption (2)**
> > > >
> > > > Crucially, in the intermediate and later stages of diffusion (where $\alpha\to 1$), this gap becomes extreme. As $\alpha\to1$, the collision probability for masking approaches 1, whereas for substitution, it approaches $(1/|\mathcal V|)^L$, where distinct graphs are highly scattered. That is, $\lim_{\alpha\to1}\frac{\Pr[C_t(g_1,g_2)]^{\mathrm{mask}}}{\Pr[C_t(g_1,g_2)]^{\mathrm{sub}}}\approx |\mathcal V|^L$.
> > > >
> > > > We visualize the gap between $\Pr[C_t(g_1,g_2)]^{\mathrm{mask}}$ and $\Pr[C_t(g_1,g_2)]^{\mathrm{sub}}$ in Fig. 8 (Appendix F) in our revised manuscript. The plot confirms that the gap becomes extreme as $\alpha\to1$. Specifically, when $\alpha=1$, the ratio $\Pr[C_t(g_1,g_2)]^{\mathrm{mask}} / \Pr[C_t(g_1,g_2)]^{\mathrm{sub}}\approx 10^{16}$. This confirms that while substitution can theoretically produce multimodal targets, the probability of such an event is orders of magnitude lower than masking with fair amount of vocabulary. Thus, state-clashing is much more pertinent to MDMs.
> > > >
> > > > To improve our claim and make the manuscript clearer, we have revised the text to reframe our argument around the multimodality of target distribution in Sec 4.1 and Appendix F, and we have included the toy example above in Appendix F, to qualitatively support our explanation.

---

> ### Comment · Reviewer_88F6 · 2025-11-27
> **I modified my score to 4**
>
> I modified my score to 4, further discussion is needed to address my remaining concerns.

---

> ### Author Response · Authors · 2025-11-30
> **Further clarification on comparison with symmetry breaking**
>
> Thank you for pointing this out and giving us the opportunity for further clarification. To this end, we provide two examples where state-clashing and symmetry breaking formulations clearly differ. Please refer to Appendix G for more details with visual examples.
>
> **State-clashing on non-symmetric intermediate states.**
>
> We point out that state-clashing can occur even for non-symmetric cases, not only covering self-symmetric inputs that the reviewer mentioned. As illustrated in Fig. 9 (a) in our revised manuscript, masking the fluorine atom and bond in 2-Fluoroethanol produces the same masked state as masking the chlorine atom and bond in 2-Chloroethanol. The collapsed masked state is not self-symmetric, yet the denoiser still faces a multimodal reconstruction target.
>
> We clarify that our emphasis is on mismatch between the joint distribution and the product of marginal distribution for denoising labels, *e.g.*, $p(x_{1}, x_{2}) \neq p(x_{1})p(x_{2})$. This can happen even when there exists no symmetry in intermediate states nor the reconstruction targets.
>
> **Symmetry breaking for non-clashing states.**
>
> In contrast, symmetry breaking can help even when there exist no state-clashing, which we depict an example in Fig. 9 (b) in our updated manuscript. Here, even when no state-clashing happens between distinct graphs, the reconstruction will fail for permutation-equivariant neural networks.
>
> Overall, we would like to clarify that (a) state-clashing and (b) symmetry breaking refers to distinct problems arising from lack of expressive power in (a) element-wise independent prediction and (b) permutation-equivariant architectures used by the decoder, respectively.

---

### Official Review · Reviewer_syMP · 2025-10-31

**Soundness:** 3
**Presentation:** 3
**Contribution:** 3
**Rating:** 6
**Confidence:** 3

**Summary:**

This paper focuses on the adaptation problem of Masked Diffusion Models (MDMs) for discrete data in molecular graph generation. It points out that "fixed, element-independent" forward masking scheduling leads to different molecules collapsing to the same intermediate state in the forward trajectory, making reverse denoising, typically unimodal and predicting independently by node or edge, difficult to learn the correct reconstruction target. To address this, the paper proposes MELD: which learns the forward masking rate at the element level (node/edge) and assigns an independent erosion trajectory to each graph element through a parameterized noise scheduling network; it is jointly optimized with the reverse denoising network during training. The authors claim that MELD achieves high efficiency in unconditional generation of QM9 and ZINC250K graphs and outperforms standard MDM and several diffusion baselines in distribution alignment and property alignment.

**Strengths:**

1.An intuitive explanation and formal analysis of the "state-clashing" phenomenon are given, pointing out that fixed, element-independent forward occlusion makes it easy for different graphs to fall into intermediate states with poor distinguishability, resulting in a highly multimodal posterior and a model approximating a "unimodal, decompositional" distribution, which in turn produces solutions with high entropy and distribution shift. Formulas (3) and (4) are relatively clear with textual explanations.

2.Both unconditional (QM9, ZINC250K, Guacamol) and conditional generation are evaluated; it also includes ablation (fixed vs. learned scheduling, node/edge/node+edge) and the "number of intermediate states" metric to characterize state-clashing.

3.The training objective employs CE weighting by node and edge, and the gradient of the discrete sampling is discussed in the pass-through estimation; these are consistent with recent MDM literature.

4.The occlusion rate is learned at the element level to avoid large-scale collisions in the middle time step, and a differentiable ST-Gumbel training path is given.

**Weaknesses:**

1.The element-level kernel renders the forward process non-equivariant, meaning the intermediate state distribution is affected by vertex permutations. For molecular graphs, this contradicts the fundamental principle that isomorphism should not alter the generative distribution. Current methods merely introduce a learnable embedding H for each graph element and "randomly permutate columns" to "distinguish graph states with the same number of nodes/edges," but this does not restore the guarantee of permutation equivariance. It needs to be proven that this forward process, which breaks equivariance, does not induce dependencies on node labels and generalization issues, especially whether relabeling input nodes during testing maintains a consistent sampling distribution.

2.The abstract and main text claim that MELD is "the first diffusion model to achieve 100% chemigenicity in unconditional generation on QM9 and ZINC250K," but several MDM baselines in Table 1 also show 100%. The wording needs to be corrected.

3.The paper does not provide an explicit collision risk function or upper and lower bound analysis; the loss in Equation (3) does not directly minimize the "collision probability". It is suggested to provide a computable proxy metric and its relationship with the gradient direction, or to supplement the appendix with a simplified derivation of the "collision probability as a function of {𝑤_{𝑖}}".

4.The manuscript states that "unless otherwise specified, standard MDM and MELD use the same DiT backbone," but were the other discrete/continuous diffusion baselines in Table 1 also retrained and had their backbones and training budgets aligned? If comparisons are only made within the MDM family without aligning the backbones/hyperparameters of external distributed models, the conclusions may overestimate the advantages of MELD. Please provide the number of training epochs, GPU configuration, total duration, and FLOPs in the appendix, as well as the retraining/reproduction practices for each baseline.

5.The use of V.U.N.↑ in Tables 3 and 6 lacks a clear explanation of its meaning and calculation in the text (it seems to be a composite score for Validity/Uniqueness/Novelty?). Please define it explicitly at its first appearance in the text.

**Questions:**

1.The statement "first 100% validity" conflicts with Table 1. It is recommended to change it to "significantly reduced FCD/NSPDK while maintaining 100% validity." Could you please report the confidence intervals for inefficiency (multiple sampling)?

2.Can a more systematic comparison be made between the key differences and complexity of existing "adaptive/category-level" scheduling (such as DiffusionBERT, GenMD4, TabDiff) and the "element-level" scheduling in this paper? Currently, only a rough comparison is made in Table 3, lacking a theoretical analysis of the differences in expressive power.

3.Please list the number of training epochs, learning rate, scheduler, backbone, number of GPUs, and training time for all baselines; and specify which baselines were retrained by the authors and which were reproduced from the original paper.

4.Please add "Node relabeling robustness test" (variance of distribution index/property MAE under multiple labels of the same molecule).

5.Table 5 only performs isomorphism counting on 12 nodes/131 samples, which is costly but has a small sample size; it is recommended to provide estimation methods for larger scales (such as approximate GI or fingerprint hash upper/lower bounds) and statistical confidence intervals.

---

> ### Author Response · Authors · 2025-11-21
> **Official Comment by Authors (1)**
>
> We sincerely appreciate the reviewer’s feedback. We would like to address the reviewer’s concerns as below.
>
> ---
>
> **[W1-1. Lack of permutation equivariance for graph generation]**
>
> Thank you for the constructive feedback. We respectfully clarify that while strict equivariance is essential for representation learning (to map isomorphic graphs to identical embeddings), graph generation only requires the learned distribution to be permutation invariant.
>
> Our method achieves this invariance not by constraining the architecture, but by marginalizing over permutations. Formally, we model the probability of a graph $G$ as the expectation over all possible node orderings $\pi$, *i.e.*, $p(G) = \sum_{\pi} p(G, \pi)$. where the summation is over the possible permutation $\pi$.
>
> Our randomized permutation strategy corresponds to maximizing the evidence lower-bound (ELBO) of this marginal log-likelihood:
> $$\log p(G) = \log \sum_{\pi} p(G,\pi) \geq \mathbb{E}_{\pi}[\log p(G | \pi)] + \text{const}.$$
>
> This stochastic symmetrization is well-established paradigm in graph generation, widely used in autoregressive models, *e.g.*, GraphRNN, GraphARM, which are inherently node-order dependent yet generate valid, invariant graph distributions.
>
> ---
>
> **[W1-2 & Q4. Node relabeling robustness test]**
>
> To further alleviate the reviewer's concern, we conduct the robustness test as follows.
>
> **Distributional Consistency**: We measure the Jensen-Shannon Divergence (JS divergence) between node and edge distribution of (1) original node ordering and (2) permuted node orderings. We conduct this experiment upon 100 different molecules sampled from the test set of ZINC250K, with 20 different permutations and upon 4 different timesteps. As can be seen, the JS divergence is remained at a consistently low level throughout timesteps, indicating the functioning of $H$ permutation technique.
>
> Table 1. JS divergence between distributions of original and permuted node orderings across different timesteps.
> | Timestep      |    0.2 |    0.4 |    0.6 |    0.8 |
> |---------------|-------:|-------:|-------:|-------:|
> | JS divergence | 0.0536 | 0.0513 | 0.0615 | 0.0777 |
>
> **Property MAE Consistency**: We measure the variance of property MAE at 4 timesteps (0.2, 0.4, 0.6, 0.8$T$), upon 20 different permutations for 111 molecules (the size of the Polymer test set). Specifically, upon a test molecule, we apply permutation, and noise it to a specific timestep $t$, and apply the full denoising process to reconstruct the molecule, and measure its property values. We take the mean of the property MAE's variance, and show it below in Table 2. It can be seen that the variance is maintained at a low level, throughout diverse timesteps, indicating equivariance is maintained at a reasonable manner.
>
> Table 2. Variance of property MAE across different timesteps.
> | Timestep  |    0.2 |    0.4 |    0.6 |    0.8 |
> |-----------|-------:|-------:|-------:|-------:|
> | SAS       | 0.0886 | 0.0582 |  0.087 |  0.051 |
> | O2 perm.  | 0.0878 |  0.048 | 0.0638 | 0.0366 |
> | N2 perm.  | 0.0826 | 0.0478 | 0.0672 | 0.0362 |
> | CO2 perm. | 0.0822 | 0.0446 | 0.0612 | 0.0422 |
>
> ---
>
> **[W2 & Q1. Clarification of statements]**
>
> Thank you for your feedback. We have revised the manuscript regarding our statement on "first 100\% validity" to improve clarity. Furthermore, following the reviewer's suggestion, we report the standard deviation of our method in Table 10 and 11 in the Appendix of our revised manuscript.
>
> ---
>
> **[W3. Minimizing collision probability]**
>
> The core idea of our approach is to assign distinct masking rates to each graph element during the forward process, thereby reducing collisions in intermediate graph states. Although the training loss (node and edge type classification) does not explicitly include a collision-risk term, we empirically observe that the learned noise schedule gradually increases the distinguishability of graph states, as shown in the following tables.
>
> Table 3. Number of unique graph states across varying timesteps in ZINC250K dataset, averaged over 3 seeds.
>
> |#(unique graphs) / Timesteps|T-100|T-75|T-50|T-25|T-1|
> |-|-|-|-|-|-|
> |Epoch=100|131 ± 0.0|131 ± 0.0|131 ± 0.0|130.7 ± 0.5|11.3 ± 0.9|
> |Epoch=200|131 ± 0.0|131 ± 0.0|131 ± 0.0|131 ± 0.0|16.0 ± 1.4|
> |Epoch=400|131 ± 0.0|131 ± 0.0|131 ± 0.0|131 ± 0.0|16.3 ± 1.2|
> |Epoch=600|131 ± 0.0|131 ± 0.0|131 ± 0.0|131 ± 0.0|17.3 ± 2.1|
>
> Table 4. Coefficient of variance (CV) of learnable embedding $H$ in varying epochs.
> |Epochs|100|200|400|600|
> |-|-|-|-|-|
> |Variance |0.013|0.041|0.153|0.885|

---

> ### Author Response · Authors · 2025-11-21
> **Official Comment by Authors (2)**
>
> **[W4 & Q3. Training details and reproduction practice of baselines]**
>
> We thank the reviewer for the helpful comment. Following common practice in prior molecular diffusion work [1, 2, 3, 4], we report official baseline results (except for GraphARM [3]) from GruM [1] for unconditional generation and GraphDiT [5] for property-conditioned generation. For GraphARM, we take the results from the original paper, since the official codebase is not available. All vanilla MDM variants (cosine, power-law, polynomial, GenMD4, DiffusionBERT, TabDiff schedules) as well as MELD were retrained by us under identical training budgets. We have added available training details and FLOPs comparison with representative baselines in Appendix C and Table 8, 9 in our revision.
>
> All of our experiments were run on a single RTX 3090 or RTX 4090 GPU per run, and MELD does not rely on increased GPU count or specialized accelerators. For reference, the GruM baseline results were obtained using RTX 3090 and RTX 2080 Ti, while GraphDiT used an A6000 GPU. The number of GPUs per run was not reported in those papers. Importantly, these configurations all fall within a similar class of commodity GPUs, and none of the compared methods (including MELD) benefits from substantially larger compute budgets or hardware advantages.
>
> ---
>
> **[W5. Clarification of V.U.N. metric]**
>
> Yes, it is a composite score for Validity, Uniqueness, and Novelty. We have added description of V.U.N. in our revised manuscript for clarity.
>
> ---
>
> **[Q2. Systemic comparison to adaptive/category-level scheduling]**
>
> To address the reviewer's request, we provide a systematic comparison based on Degrees of Freedom (DoF) to demonstrate why adaptive/category-level scheduling is theoretically insufficient to resolve state-clashing. In particular, the degree of freedom in scheduling for the adaptive/category-level scheduling depends on the number of atom types on the molecular graph being corrupted by the forward diffusion process, which is typically four to eleven. In comparison, the degree of freedom in our scheduling depends on the number of atoms, which ranges from ten to fifty.
>
> The limited DoF in category-level scheduling imposes the constraint that identical atom types should share the schedule. In a molecule consisting entirely of carbon atoms and single bonds (*e.g.*, Cyclohexane), category-level scheduling effectively collapses to a fixed scalar schedule (DoF=1). It cannot structurally distinguish one carbon from another. Consequently, this inability to break symmetries makes "state-clashing" (as illustrated in Figure 1) unavoidable for symmetric structures. MELD utilizes its instance-wise DoF to break these symmetries, which is necessary for avoiding collapse in masked graph diffusion.
>
> ---
>
> **[Q5. State-clashing on larger sample size]**
>
> We present an additional experiment to evaluate state-clashing effects on larger molecules from the large-scale Guacamol benchmark. Running full or hashed Weisfeiler-Lehman tests in graph isomorphism checks with over 12 nodes was prohibitively expensive in practice. To make the analysis tractable, we instead adopt an approximate fingerprint-based hashing method as an alternative.
>
> For each graph, we construct a canonical graph fingerprint using a set of graph properties that remain unchanged under node permutations. The fingerprint aggregates several graph descriptors, including the numbers of nodes and edges, sorted list of node degrees, and sorted multisets of node and edge types (including masked tokens). These descriptors are then concatenated into a string (*e.g.*, ``n4|e3|d:1,1,2,2|v:0,6,6,8|eattr:1,1,2``), which is then hashed using SHA-1. Graphs with identical hashes are treated as duplicates.
>
> We sample molecules from Guacamol using the maximum graph size in its test set that has sufficient number of samples (>100 samples), yielding 144 molecules containing 58 nodes each. As shown in the following table,  MELD still produces substantially more unique graphs at the last timestep compared to other MDM baselines.
>
> Table 5. Number of unique graph states across varying timesteps in Guacamol, averaged over 3 seeds.
>
> |#(unique graphs) / Timesteps|T-100|T-75|T-50|T-25|T-1|
> |-|-|-|-|-|-|
> |MDM w/ cosine|**144 ± 0.0**|**144 ± 0.0**|**144 ± 0.0**|134.3 ± 2.1|4.7 ± 1.2|
> |MDM w/ polynomial|**144 ± 0.0**|**144 ± 0.0**|**144 ± 0.0**|**144 ± 0.0**|99.3 ± 1.9|
> |MDM w/ power-law|**144 ± 0.0**|**144 ± 0.0**|**144 ± 0.0**|**144 ± 0.0**|85.7 ± 2.1|
> |MELD|**144 ± 0.0**|**144 ± 0.0**|**144 ± 0.0**|**144 ± 0.0**|**115.7 ± 3.1**|
>
> **Reference**
>
> ---
>
> [1] Graph generation with diffusion mixture, ICML '24.
>
> [2] Score-based Generative Modeling of Graphs via the System of Stochastic Differential Equations, ICML '22.
>
> [3] Autoregressive Diffusion Model for Graph Generation, ICML '23.
>
> [4] Discrete-state Continuous-time Diffusion for Graph Generation, NeurIPS '24.
>
> [5] Graph Diffusion Transformers for Multi-Conditional Molecular Generation, NeurIPS '24.

---

### Official Review · Reviewer_qGV2 · 2025-11-01

**Soundness:** 3
**Presentation:** 3
**Contribution:** 2
**Rating:** 4
**Confidence:** 5

**Summary:**

This paper investigates the application of Masked Diffusion Models (MDMs) to molecular generation. The authors identify a key limitation in standard MDMs, which they term the "state-clashing problem," where fixed, element-agnostic noise schedules cause distinct molecular graphs to collapse into identical corrupted states during the forward process. To address this, the paper proposes Masked Element-wise Learnable Diffusion (MELD), a framework that learns a flexible, per-element (atom and bond) noise schedule. Through extensive experiments on unconditional and property-conditioned molecular generation tasks, the authors demonstrate that MELD achieves 100% chemical validity on QM9 and ZINC250K and outperforms standard MDMs and other diffusion-based baselines in distributional and property alignment.

**Strengths:**

1. Originality and Significance. The paper makes a significant and original contribution by identifying the "state-clashing problem" as an obstacle to applying standard MDMs to structured data like molecular graphs. The core idea of learning an element-wise forward process to orchestrate distinct corruption trajectories is an elegant and insightful solution.
2. Quality. The technical quality of the work is high. The hypothesis about state-clashing is well-motivated and convincingly demonstrated through both theoretical formalization and empirical analysis. The proposed MELD framework is a technically sound and well-designed solution. The experimental evaluation is comprehensive, covering multiple datasets (QM9, ZINC250K, Polymers, Guacamol) and tasks (unconditional, conditional), and benchmarking against a wide range of strong baselines.
3. Clarity. The paper is well-written and easy to follow. The state-clashing problem is introduced with intuition and supported by clear illustrations. The experiments are well-structured, and the results are clearly communicated through tables and figures.

**Weaknesses:**

1. The paper's central claim of superiority is undermined by an incomplete set of baseline comparisons. While MELD is shown to be effective against standard MDMs and some diffusion models, it omits a direct comparison to some relevant works. Methods presented in "Conditional Diffusion Based on Discrete Graph Structures for Molecular Graph Generation" and "Learning Joint 2-D and 3-D Graph Diffusion Models for Complete Molecule Generation" have demonstrated exceptional performance on ZINC250K benchmarks. These models achieve their results using strong denoising architectures but with simple, fixed noise schedules.
This raises a critical question: Is the added complexity of learning the forward process truly necessary if a stronger denoising architecture with a fixed schedule can achieve similar results? The paper needs to more explicitly articulate the unique advantages of its approach beyond incremental performance gains. For instance, does MELD offer better parameter efficiency or faster training? Without a clear, compelling advantage, the rationale for introducing a learnable noise schedule, which slows down training, is weakened.

2. Ineffective Visualization in Figure 3. The visualization in Figure 3, which aims to demonstrate MELD's faster recovery during the reverse process, is not very effective. The overlaid text ("[MASK]") and molecular structures are small, cluttered, and difficult to parse. This makes it challenging to visually verify the claim that MELD reconstructs meaningful fragments earlier than the element-agnostic schedule. A clearer visualization would be much more impactful.

3. Missing Discussion of Training Overhead in Main Text. The paper introduces a learnable noise scheduling network and a joint optimization procedure, which inherently adds computational cost and complexity during training compared to models with a fixed forward process. However, the main text lacks any discussion of this overhead.  A summary of these findings should be included in the main paper to provide readers with a complete picture.

**Questions:**

1. To better situate your work, could you discuss and ideally provide an experimental comparison against more baselines with a fixed schedule?  This is essential to demonstrate the advantages of a learnable schedule over a powerful denoiser with a fixed schedule.
2. Could you revise Figure 3 for better clarity? The current visualization is difficult to interpret. A clearer format, such as using 2D renderings and highlighting unmasked elements, would more effectively demonstrate the claimed faster recovery of your method.
3. Could you add a brief discussion of the training overhead (e.g., increased training time) introduced by the learnable scheduling network to the main paper? This would help readers better understand the practical trade-offs of your approach.

---

> ### Author Response · Authors · 2025-11-21
> **Official Comment by Authors (1)**
>
> We sincerely appreciate the reviewer’s feedback. We would like to address the reviewer’s concerns as below.
>
> ---
>
> **[W1-1. Necessity and complexity of learnable schedule vs. expressive architectures]**
>
> We demonstrate that the learnable schedule is not merely an alternative to a stronger architecture, but a necessity for resolving the state-clashing problem inherent to masked diffusion models (MDMs).
>
> First, our experiments already show that expressive architectures are not sufficient to resolve state-clashing in MDMs. In Table 1 in our main manuscript, the MDM baselines and our MELD utilize the exact same expressive DiT (Diffusion Transformer) backbone. The catastrophic failure of MDM baselines in the table validates the necessity of our learnable schedule scheme.
>
> Next, we show that the "added complexity" is computationally insignificant. As shown in Table 13 in the Appendix, MELD adds only $\sim$ 0.01M parameters and negligeable impact on inference latency (an overhead of only up to 0.032 seconds across molecule sizes).
>
> Finally, we note the significance of MELD apart from the performance gains. MELD is the first successful adoption of MDMs, which is the paradigm behind state-of-the-art image and language models, to molecular graphs. By solving the unique modality-specific failure mode (state-clashing), MELD unlocks the scalability and parallel generation benefits of MDMs for the graph domain, offering a foundation for future research that standard discrete diffusion cannot provide.
>
> ---
>
> **[W1-2. Comparison with CDGS & JODO]**
>
> Thank you for introducing important baselines, which we will reference in our future manuscript. We present the performance comparison with MELD, CDGS [1], and JODO [2] in Table 1. Please note that the reproduced baseline performance differs from those in the original papers due to differences in formal charge assignment and bond order handling within the graph-to-molecule conversion pipeline, which we adopt from GruM and GDSS.
>
> Table 1. Performance and efficiency comparison between MELD and baselines on ZINC250K dataset.
>
> | Methods | Total iterations | Diffusion steps | Valid.    | FCD      | NSPDK      | Scaf.     | Uniq.      | Novel.    | WD/logP   | WD/QED    | WD/SA     |
> | ------- | ---------------- | --------------- | --------- | -------- | ---------- | --------- | ---------- | --------- | --------- | --------- | --------- |
> | CDGS    | 2000000          | 1000            | **100.0** | 2.41     | 0.0007     | 0.4747    | 99.97      | **99.99** | **0.220** | 0.008     | 0.292     |
> | JODO    | 1500000          | 1000            | **100.0** | 1.58     | **0.0004** | **0.599** | **100.00** | 99.94     | 0.464     | 0.022     | 0.380     |
> | MELD    | **263400**       | **500**         | **100.0** | **1.51** | 0.0008     | 0.560     | 99.99      | 99.95     | 0.340     | **0.004** | **0.110** |
>
> In Table 1, one can observe that MELD performs better or similar to CDGS for  most of the metrics except for Wasserstein distance for logP scores (WD/logP). The performance is comparable to JODO, with better scores for FCD, WD/logP, WD/QED, WD/SA, worse scores for NSPDK, Scaf, and similar scores for Valid. Uniq. and Novel.
>
> Importantly, MELD attains this performance with significantly fewer total training iterations (~7.6× reduction compared to CDGS; ~5.7× compared to JODO) and with half the diffusion sampling steps, highlighting the efficiency of our method.
>
> ---
>
> **[W2 & Q2. Ineffective visualization of Fig. 3]**
>
> Thank you for your feedback. We have revised Fig. 3 to improve the visualization, in the updated version of our manuscript.
>
> ---
>
> **[W3 & Q3. On training overhead]**
>
> Thank you for the feedback. To address the concerns, we've conducted a computational overhead of MELD, showing that it introduces only negligible computational and memory overhead. We report the (1) FLOPs, (2) execution time and (3) peak memory when processing a molecule size of $|V|=100$ with a batch size of 32. All experiments were conducted on an NVIDIA GeForce RTX 4090 GPU and an AMD EPYC 7K62 48-Core Processor.
>
> Table 2. Computation and memory overhead of MELD and vanilla MDMs with varying noise schedules.
> |                  | Fixed Polynomial | Fixed Cosine | TabDiff      | MELD        |
> |------------------|------------------|--------------|--------------|-------------|
> | FLOPs (GMac)     |           315.23 |       315.23 |        318.5 |       318.5 |
> | Execution (s)    |     0.9443941116 | 0.9398224354 | 0.9510345459 | 1.156838417 |
> | Peak Memory (MB) |          2810.56 |      2810.56 |      3352.12 |      2840.5 |

---

> > ### Author Response · Authors · 2025-11-21
> > **Official Comment by Authors (2)**
> >
> > **[Q1. Noise scheduling comparison with different architectures]**
> >
> > We have included comparisons against additional fixed-noise–scheduling baselines (CDGS and JODO) in **[W1]**. In addition, we evaluate whether our method shows consistent improvement across different denoiser architectures beyond DiT. To this end, we replace the DiT backbone with the graph transformer architecture used in DiGress, and assess the performance of both MELD and vanilla MDMs on the ZINC250K dataset.
> > Note that we do not incorporate the additional spectral features used in DiGress across all settings.
> >
> > As shown in Table 3, MELD continues to outperform all fixed-schedule MDM variants by a substantial margin, even with different denoiser backbone. This demonstrates that MELD’s effectiveness is not tied to a specific model backbone.
> >
> > Table 3. Performance comparison of MELD against vanilla MDM variants (fixed noise schedules) on ZINC250K dataset, with graph transformer of DiGress as denoiser architecture. Note that we do not incorporate the additional spectral features used in DiGress.
> > | Methods + DiGress Backbone            | Valid.    | FCD      | NSPDK      | Scaf.     | Uniq.      | Novel.    |
> > | ------------------ | --------- | -------- | ---------- | --------- | ---------- | --------- |
> > | MDM (w/ cosine)    | 99.88     | 44.52    | 0.5073     | 0.0000    | 2.40       | **100.0** |
> > | MDM (w/ power-law) | 99.99     | 42.97    | 0.3777     | 0.0000    | 5.38       | **100.0** |
> > | MDM (w/ polynomial)| 98.95     | 40.78    | 0.3871     | 0.0000    | 5.95       | **100.0** |
> > | MELD               | **100.0** | **4.64** | **0.0044** | **0.370** | **99.99**  | **100.0** |
> >
> > **Reference**
> >
> > ---
> >
> > [1] Conditional Diffusion Based on Discrete Graph Structures for Molecular Graph Generation, AAAI '23.
> >
> > [2] Learning Joint 2D & 3D Diffusion Models for Complete Molecule Generation, arXiv '23.

---

### Official Review · Reviewer_HAmM · 2025-11-08

**Soundness:** 2
**Presentation:** 3
**Contribution:** 2
**Rating:** 6
**Confidence:** 3

**Summary:**

This paper proposes a state-clashing problem, meaning that when masked diffusion models are applied to molecular generation, different molecules collapse into the same state, which increases the difficulty of learning. The authors propose an element-wise learnable method that alleviates this issue by learning different corruption rates.

**Strengths:**

1.	This paper explores the performance of the currently popular MDM in the field of molecular generation, which is a research topic worth pursuing.
2.	The authors make improvements based on the MDM by introducing element-wise embedding to adapt it to molecular generation tasks.
3.	The authors also validate the effectiveness of the method on large-scale datasets such as Guacamol.

**Weaknesses:**

1.	Some overclaims in the paper need clarification, such as the statement that in previous work the transition probabilities between elements in the forward process are all uniformly distributed.
2.	When proposing the state-clashing problem, the authors lack demonstrations on large-scale datasets. This makes it difficult to convince readers whether such a problem truly exists.
3.	The cases in Figure 2 are not easy to understand and require clearer explanation.
4.	Some parts of the method that are based on existing methods could be moved to the appendix.

**Questions:**

1.	The authors propose an element-wise learnable method that learns different corruption rates. Is there any suitable case study that has been analyzed to show the relationship between the learned different rates and the various types of elements?

---

> ### Author Response · Authors · 2025-11-21
> **Official Comment by Authors**
>
> We sincerely appreciate the reviewer’s feedback. We would like to address the reviewer’s concerns as below.
>
> ---
>
> **[W1. Clarification of statements on transition probabilities]**
>
> Thank you for the comment. We have revised the manuscript to clarify this point. We now state that previous molecular diffusion models use a fixed, *element-agnostic* transition probability.
>
> ---
>
> **[W2. Demonstration of state-clashing on large-scale dataset]**
>
> To alleviate your concerns, we provide demonstration of state-clashing on the large-scale Guacamol dataset with ~1.3M molecules. To this end, we quantify the degree of state clashing for different noise schedules of the forward masked diffusion process. Since direct graph isomorphism tests are computationally infeasible to quantify the degree of state-clashing, we use approximate fingerprint-based hashing method to count the number of graphs that are provably distinct.
>
> Table 1. Number of graphs with distinct fingerprints across varying timesteps in the Guacamol dataset, averaged over 3 seeds.
>
> |#(unique fingerprints) / Timesteps|T-100|T-75|T-50|T-25|T-1|
> |-|-|-|-|-|-|
> |cosine|**144 ± 0.0**|**144 ± 0.0**|**144 ± 0.0**|134.3 ± 2.1|4.7 ± 1.2|
> |polynomial|**144 ± 0.0**|**144 ± 0.0**|**144 ± 0.0**|**144 ± 0.0**|99.3 ± 1.9|
> |power-law|**144 ± 0.0**|**144 ± 0.0**|**144 ± 0.0**|**144 ± 0.0**|85.7 ± 2.1|
> |MELD|**144 ± 0.0**|**144 ± 0.0**|**144 ± 0.0**|**144 ± 0.0**|**115.7 ± 3.1**|
>
> In Table 1, one can observe that our MELD scheduling results in a larger number of unique fingerprints at each time-step. We provide more experimental details in Appendix D.6 in our revised manuscript.
>
> ---
>
> **[W3 & 4. Upon Fig. 2 and manuscript coordination]**
>
> Thank you for the feedback. We revise the clarity of our explanation on Fig. 2 in our revised manuscript.
>
> To clarify, Fig. 2 visualizes the denoiser's prediction entropy when reconstructing masked bonds in the given molecules. In the first two rows, we mask all nitrogen–carbon bonds in o- and m-phenylenediamine. Because masking removes the distinguishing nitrogen atoms, both molecules collapse into an identical symmetric benzene backbone, creating a strong state-clashing scenario. Under an element-agnostic schedule, the denoiser therefore exhibits higher entropy when predicting the masked bond types, as many distinct underlying configurations can map to the same corrupted state. In contrast, the denoiser trained with MELD's element-wise schedule produces much lower entropy.
>
> In the third row (2-chloro-4-fluorotoluene), masking only the chlorine–carbon bond creates asymmetric molecule, so state-clashing is intrinsically less severe compared to benzene ring in the previous case. As a result, even the denoiser trained with an element-agnostic schedule shows relatively low entropy.
>
> Additionally, as the reviewer suggested, we have reorganized our manuscript by moving detailed STGC training into the Appendix and simplifying the preliminaries of molecular diffusion models that overlap with existing literature.
>
> ---
>
> **[Q1. Relationship between learned corruption rates and element types]**
>
> We clarify that the learned corruption rates are not related to specific element types by design. Instead, MELD assign distinct masking rate based on the position of each element. Note that this breaks permutation, but we randomly shuffle to columns to prevent the noise schedule from overfitting to specific element indices.
>
> Since MELD is not designed to be type-dependent, our analysis focuses instead on how the learned schedules vary across element-wise positions. In Appendix D.3, we examine the per-step variation of node and edge masking rates and observe that edges consistently exhibit larger variance than nodes. This suggests that the model learns to distinguish edges more aggressively. We believe this behavior naturally arises from the structure of molecular graphs. Many distinct molecules share the same atom types but differ in bond configurations (*e.g.*, structural isomers), whereas two molecules that share the same connectivity but differ only in atom labels are far less common due to valency constraints.

---

### Author Response · Authors · 2025-11-30
**Global Response**

We sincerely thank the reviewers for their constructive feedback and the AC for coordinating the process.

During the discussion period, we actively engaged with reviewers, providing detailed theoretical derivations, quantitative validation on state-clashing, computational overhead analysis, and extensive baseline comparisons. We note that Reviewer 88F6, the only reviewer, who participated in the discussion, has recently raised the score from 2 to 4, and expressed willingness to continue further discussion to resolve any remaining concerns. We appreciate this reconsideration and have provided additional response to the reviewer’s follow-up questions.

**Key Revisions and Clarifications**

We have improved the manuscript to revise our claims and to incorporate comparison with symmetry breaking and with substitution-based corruption. Moreover, we have carefully addressed every issue raised in the initial reviews through additional experiments, comprehensive clarifications, and new baseline comparisons. The key improvements are summarized below:

**1. Theoretical & Empirical Validation of State-Clashing (Reviewers HAmM, syMP, 88F6)**

- **Large-Scale Verification (Reviewers HAmM, syMP):** To prove state-clashing exists beyond small datasets, we conducted experiments on the large-scale Guacamol dataset. Using fingerprint-based hashing, we showed MELD maintains significantly higher structural diversity (more unique fingerprints) at intermediate timesteps compared to cosine, polynomial, and power-law schedules.
- **State-clashing comparison with substitution-based corruption (Reviewer 88F6):** We provided a rigorous mathematical derivation comparing masked vs. substitution-based corruption. We demonstrated that the probability of state-clashing in substitution-based corruption is orders of magnitude lower than masking with a realistic amount of vocabulary, proving that state-clashing is pertinent to MDMs that substitution methods avoid.
- **Distinction of state-clashing from symmetry breaking (Reviewer 88F6):** We presented a systematic comparison between state-clashing and symmetry breaking, showing that two problems arise from different causes (element-wise independent prediction and permutation-equivariant denoiser, respectively).

**2. Comparisons to Stronger Baselines & Architecture Independence (Reviewers qGV2, 88F6)**

- **New Baselines:** We added direct comparisons against CDGS and JODO (Reviewer qGV2) and SymPE (Reviewer 88F6). MELD achieves comparable performance with the strong baseline JODO with significantly greater training efficiency (~7.6x fewer iterations). Furthermore, it surpasses symmetry breaking approach, SymPE, across all metrics.
- **Architecture Independence:** To prove MELD’s gains are not solely due to the DiT backbone, we applied MELD to the DiGress (graph transformer) backbone. MELD still outperformed fixed-schedule MDMs by a large margin, confirming the necessity of element-wise learnable scheduling regardless of the denoising architecture.

**3. Robustness and Equivariance Analysis (Reviewer syMP)**

- We clarified that while our element-wise schedule breaks strict permutation equivariance, it maintains a valid generative distribution via marginalization. We supported this with a **node relabeling robustness test**, showing that the Jensen-Shannon divergence and Property MAE variance remain consistently low across permutations, ensuring stable generation.

**4. Computational Overhead & Visualizations (Reviewers HAmM, qGV2)**

- **Efficiency:** We added a computational overhead analysis comparing MELD with other diffusion models (GruM, DiT, DiGress) and with advanced noise scheduling methods. The results show that MELD incurs comparable overhead to existing approaches.
- **Visual Clarity:** We revised Figure 3 to more clearly illustrate faster fragment reconstruction achieved by MELD compared to element-agnostic schedule. Additionally, we add a clear explanation of Figure 2 in Section 4.1 in our revised manuscript.

We believe these enhancements highlight the significance of our problem formulation and effectiveness of MELD.

Thank you once more for your efforts in reviewing our paper.

Sincerely,

The Authors

---

### Meta-Review · Area_Chair_Vbnf · 2025-12-19

**Summary:**

This paper investigates the issue of state-clashing in the application of masked diffusion models to molecular generation, proposing an element-wise learnable method to mitigate this problem. Experiments conducted on the QM9 and ZINC datasets demonstrate the effectiveness of the proposed approach. While reviewers acknowledge the work's originality and significance, several major concerns have been raised, as outlined below:

1. Clarity and Presentation: The manuscript requires improvements in clarity, including addressing certain overclaims, providing more detailed case studies in figures/tables, and enhancing visualization for better interpretability.

2. Limited Data Scale: The experimental evaluation lacks demonstration on large-scale datasets, which is crucial for assessing real-world applicability.

3. Connection to Symmetry Breaking: The relationship to symmetry-breaking literature is neither discussed nor compared, leaving a gap in contextualizing the proposed method.

4. Comparison with Substitution-Based Corruption: The work does not include comparisons with substitution-based corruption methods, which are relevant alternatives.

5. Potential Technical Issue: The introduced technique may inadvertently introduce non-equivariance, posing a risk to the model's theoretical consistency.

6. Theoretical Analysis: A more rigorous theoretical foundation is needed, such as an explicit formulation of collision risk or bound analysis.

7. Comparison Fairness: Some comparisons appear unfair due to unaligned backbones or hyperparameters with baseline models, which may skew results.

8. Experimental Evaluation: The evaluation is insufficient; more baseline models should be included, and an efficiency analysis should be conducted.

9. Computational Cost Analysis: Further analysis of computational cost and complexity is necessary to assess scalability and practical feasibility.

**Reviewer Concerns:**

Most of the concerns have been well addressed in the rebuttal.

**Reviewer Scores:**

The reviewer, originally scoring the paper a 2, indicated during the discussion a willingness to raise the score to 4, and further if his remaining concerns were fully addressed. Given that the authors' responses have been detailed and persuasive in resolving these points, it is reasonable to anticipate that this reviewer would now be supportive of a weak accept (score of 6).

---

### Decision · Program_Chairs · 2026-01-26

Accept (Poster)